# The OsSGS3-tasiRNA-OsARF3 module orchestrates abiotic-biotic stress response trade-off in rice

Xueting Gu [1,8], Fuyan Si[2,8], Zhengxiang Feng[3,8], Shunjie Li[3,8], Di Liang[1,8], Pei Yang[3], Chao Yang[2], Bin Yan[2], Jun Tang[1], Yu Yang[3], Tai Li[3], Lin Li[3], Jinling Zhou[3], Ji Li[2], Lili Feng[1], Ji-Yun Liu[1], Yuanzhu Yang[4], Yiwen Deng[1], Xu Na Wu[3], Zhigang Zhao[5], Jianmin Wan[5], Xiaofeng Cao [2,6,7], Xianwei Song [2] ✉, Zuhua He [1] ✉ & Junzhong Liu [3] ✉

Recurrent heat stress and pathogen invasion seriously threaten crop production, and abiotic stress often antagonizes biotic stress response against pathogens. However, the molecular mechanisms of trade-offs between thermotolerance and defense remain obscure. Here, we identify a rice thermosensitive mutant that displays a defect in floret development under high temperature with a mutation in S*UPPRESSOR OF GENE SILENCING 3a* (*OsSGS3a*). OsSGS3a interacts with its homolog OsSGS3b and modulates the biogenesis of *trans*-acting small interfering RNA (tasiRNA) targeting *AUXIN RESPONSE FACTORS* (*ARFs*). We find that OsSGS3a/b positively, while OsARF3a/b and OsARF3la/lb negatively modulate thermotolerance. Moreover, OsSGS3a negatively, while OsARF3a/b and OsARF3la/lb positively regulate disease resistance to the bacterial pathogen *Xanthomonas oryzae* pv. *oryzae* (*Xoo*) and the fungal pathogen *Magnaporthe oryzae* (*M. oryzae*). Taken together, our study uncovers a previously unknown trade-off mechanism that regulates distinct immunity and thermotolerance through the OsSGS3-tasiRNA-OsARF3 module, highlighting the regulation of abiotic-biotic stress response trade-off in plants.

Frequent heat waves are one of major abiotic stresses associated with climate change, which not only directly affects plant growth and development but also often compromises its immunity against pathogen[1–5]. For the modern agriculture, breeding crops with multiple biotic and abiotic resistance/tolerance is the most efficient approach to safeguard food production in the ever-changing climate. Basically, plants must adjust their prioritization towards defense or stress tolerance to survive in adverse environments. In most cases, exposure of plants to abiotic stresses, such as heat, cold, drought, and salinity, often has a negative effect on defense responses[6]. Heat

[1]National Key Laboratory of Plant Molecular Genetics, CAS Center for Excellence in Molecular Plant Sciences, Institute of Plant Physiology and Ecology, Chinese Academy of Sciences, 200032 Shanghai, China. [2]State Key Laboratory of Plant Genomics and National Center for Plant Gene Research, Institute of Genetics and Developmental Biology, Chinese Academy of Sciences, 100101 Beijing, China. [3]Center for Life Sciences, School of Life Sciences, State Key Laboratory of Conservation and Utilization of Bio-Resources in Yunnan, Yunnan University, 650500 Kunming, China. [4]Department of Rice Breeding, Hunan Yahua Seed Scientific Research Institute, 410119 Changsha, Hunan, China. [5]National Key Laboratory for Crop Genetics and Germplasm Enhancement, Nanjing Agricultural University, 210095 Nanjing, China. [6]University of Chinese Academy of Sciences, 100039 Beijing, China. [7]CAS Center for Excellence in Molecular Plant Sciences, Chinese Academy of Sciences, 100101 Beijing, China. [8]These authors contributed equally: Xueting Gu, Fuyan Si, Zhengxiang Feng, Shunjie Li, Di Liang. ✉e-mail: xwsong@genetics.ac.cn; zhhe@cemps.ac.cn; liujunzhong@ynu.edu.cn

stress not only suppresses immune responses including cytosolic $Ca^{2+}$ influx and reactive oxygen species (ROS) production[7–9], but also suppresses effector-triggered immunity (ETI) through modulating the activity of nucleotide-binding, leucine-rich repeat (NLR) immune receptors[10].

Plant growth-defense trade-offs are often regulated by plant hormones and transcription factors[11]. However, our knowledge regarding how plants coordinate biotic and abiotic stress responses remains rather limited[6,12]. It was reported that heat inhibits the production and key signaling modules of salicylic acid (SA), a central plant defense hormone[3,4,13]. In pepper (*Capsicum annuum*), the NAC-type transcription factor NAC DOMAIN CONTAINING PROTEIN 2c (CaNAC2c) coordinately regulates immunity and thermotolerance through its differential subcellular localization and interacting proteins[14]. A chromatin remodeling modulator CaSWC4 modulates the immunity-thermotolerance trade-off by recruiting transcription factors CabZIP63 and CaWRKY40[15]. AVRPPHB SUSCEPTIBLE 3 (PBS3), an acyl-adenylate/thioester-forming enzyme, balances plant immunity, salt stress responses and phyllosphere bacterial communities to maintain plant fitness through abscisic acid (ABA)-SA crosstalk under combined biotic and abiotic stresses[16]. The transcription factor XYLEM NAC DOMAIN 1 (XND1) in *Arabidopsis*, and SlERF84 and SR/CAMTA transcription factors SlSR1/3L/1L in tomato (*Solanum lycopersicum*, *Sl*) modulate the trade-off between abiotic and biotic stress responses[17–19]. Interestingly, the immune receptor, NLR locus NLR4/ACQUIRED OSMOTOLERANCE (ACQOS), negatively modulates plant abiotic stress responses[20], while some immune signaling components, such as MITOGEN-ACTIVATED PROTEIN KINASE 5 (OsMAPK5) and ACTIVATED DISEASE RESISTANCE 1 (ADR1) enhance abiotic stress tolerance[21,22]. These studies reveal the complicated intersections between biotic and abiotic stress responses. However, how heat affects plant–biotic interaction in crops remains largely unclear.

Small RNAs are a class of noncoding RNAs that function as key guide molecules to regulate gene expression and maintain genome stability in almost all biological processes in plants[23–25]. Phased small-interfering RNAs (phasiRNAs) are a specific type of secondary siRNAs that are generated from phasiRNA-generating (*PHAS*) precursor transcripts in a phased manner with an interval of 21- or 24-nucleotides (nt)[26]. Some phasiRNAs such as *trans*-acting small-interfering RNAs (tasiRNAs) function in modulating the expression of target mRNAs or noncoding transcripts[26–28]. In *Arabidopsis*, *TAS* transcripts are targeted and cleaved by miR173-Argonaute 1 (AGO1)-containing RNA-induced silencing complex (RISC) or miR390-AGO7-containing RISC[28,29]. RNA-DEPENDENT RNA POLYMERASE 6 (RDR6) is recruited to the cleaved fragments with the aid of a putative RNA export protein SILENCING DEFECTIVE 5 (SDE5) and a dsRNA-binding protein SUPPRESSOR OF GENE SILENCING 3 (SGS3)[30,31]. The cleaved fragments are converted into double-stranded RNAs (dsRNAs) by RDR6 and then processed by Dicer-like (DCL) proteins into tasiRNAs, which modulate the expression of corresponding targets such as *MYBs*, *HEAT-INDUCED TAS1 TARGETS* (*HTTs*), and *AUXIN RESPONSE FACTORS* (*ARFs*)[28,29,32]. Some phasiRNAs have been predicted to target large or conserved gene families, such as NLR disease resistance genes, which may fine-tune plant immune system[33,34]. The crucial regulatory roles of phasiRNAs in plant growth and development have been well reviewed[26], while our current understanding of the roles of phasiRNAs in plant abiotic and biotic stress responses as well as their trade-off remains scarce.

We previously reported that heat-induced transgenerational degradation of SGS3 protein compromised the biogenesis of tasiRNAs, leading to the release of transgene-mediated gene silencing, early flowering, and attenuated immunity in *Arabidopsis*[35,36]. OsSGS3a also interacts with the p2 protein of *Rice stripe virus* (*RSV*)[37]. These findings suggest that SGS3 may coordinate the trade-off between heat stress and immunity. In this study, we identify a temperature-sensitive

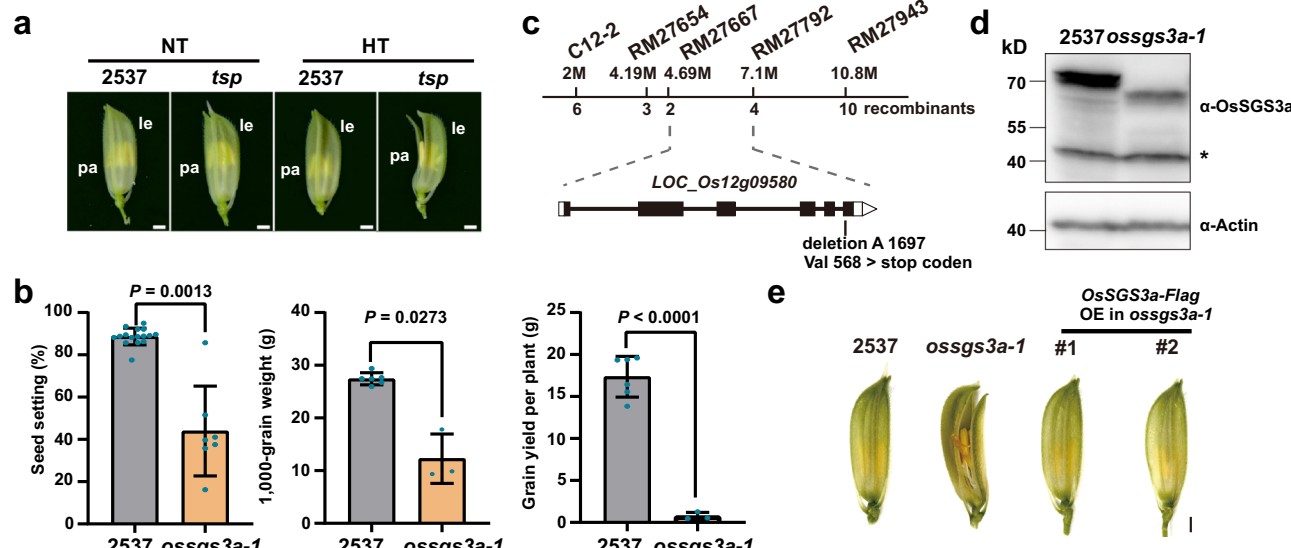

**Fig. 1 | Isolation and characterization of a thermosensitive *ossgs3a* mutant.**
**a** The glumes of thermosensitive abnormal *palea* (*tsp*) mutant and 2537 (wild-type) in Shanghai in the summer with high field temperature (HT, ≥35 °C) frequently occurring at rice booting stage, or in Hainan in the winter with normal temperatures (NT) suitable for rice growth. Pa, palea; le, lemma. **b** The seed-setting rates, 1000-grain weight, and grain yield per plant of *tsp* and 2537 grown under high field temperature. Values are mean ± s.d. (*n* = number of biologically independent samples in the graph). Significant differences were determined by two-tailed Student's *t* test for pairwise comparisons. **c** Map-based gene cloning of *OsSGS3a*. The locus responsible for the mutant phenotype was preliminarily mapped to a region between markers RM27667 and RM27792. Combining with the whole-genome sequencing-based MutMap method, a single nucleotide deletion at position 1697 in the last exon of *OsSGS3a* in *tsp* was identified. The chromosomal locations of markers and the numbers of recombinants are indicated by the numbers above and below the chromosome, respectively. **d** Western blotting analysis of OsSGS3a protein levels using anti-OsSGS3a antibody. OsSGS3a protein (69.4 kD) was detected in 2537 while the truncated OsSGS3a protein (64.6 kD) was produced in *tsp*. Equal protein loading was confirmed with the anti-OsActin antibody. The star indicates unspecific bands. Experiments were repeated three times with similar results (**b**, **d**). **e** T₁ transgenic plants expressing the Flag-tagged coding sequence of *LOC_Os12g09580* (*OsSGS3a*) developed glumes that were morphologically identical to 2537 under high field temperature. Scale bars, 1 mm (**a**, **e**). Source data are provided as a Source Data file (**b**, **d**).

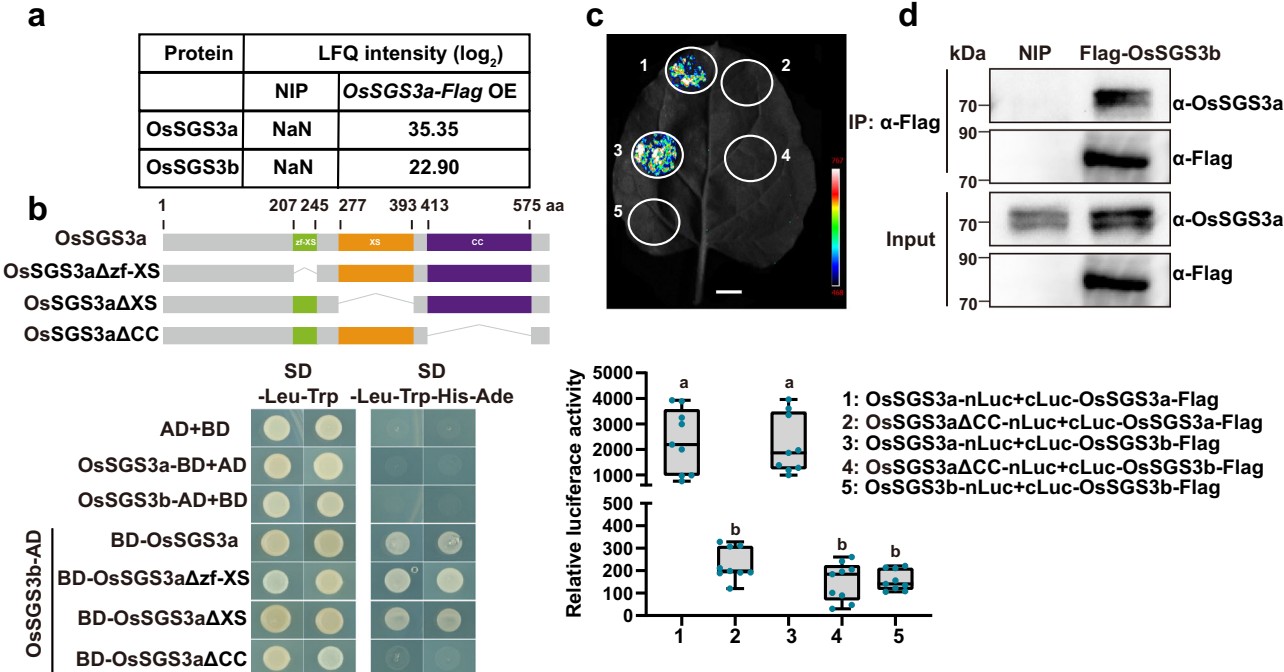

**Fig. 2 | OsSGS3a interacts with OsSGS3b. a** The abundance of OsSGS3a and OsSGS3b in the IP-MS samples from OsSGS3a-Flag transgenic plants and NIP grown in the growth chamber at 28 °C. **b** OsSGS3a interacts with OsSGS3b and the CC domain of OsSGS3a is essential for their interaction in yeast. The different domains of OsSGS3a protein are indicated. **c** Split luciferase complementation (SLC) confirmation of the OsSGS3a and OsSGS3a/OsSGS3b interactions in *N. benthamiana*. Fluorescence signal intensity is indicated. Relative luciferase activity of the indicated protein-protein interactions was measured, shown as box plots ($n = 9$, biologically independent samples). Box plots show the median (central line) and interquartile range (IQR; from the 25th to 75th percentile); whiskers extend to minimum and maximum values within 1.5 times the IQR. Lowercase letters indicate significant differences at $P < 0.05$, as determined by one-way ANOVA with the Tukey's HSD post hoc test. Exact $P$ values are provided in Supplementary Data 7. **d** Detection of OsSGS3a interaction with OsSGS3b by Co-immunoprecipitation (Co-IP) in rice. Source data are provided as a Source Data file (**c**, **d**). Experiments were repeated two times with similar results (**b**–**d**).

*ossgs3a-1* mutant. Deep sequencing of small RNAs reveals that OsSGS3a is required for the biogenesis of tasiRNAs. We further identify OsSGS3a-dependent tasiRNA-*OsARF3s* as a crucial module responsible for the trade-off between thermotolerance and immunity, which negatively regulates disease resistance to both bacterial *Xanthomonas oryzae* pv. *oryzae* (*Xoo*) and fungal *Magnaporthe oryzae* (*M. oryzae*), but positively modulates thermotolerance in rice. Thus, our results reveal that the OsSGS3-tasiRNA-OsARF3 module coordinates the trade-off between abiotic and biotic stress responses.

## Results

### Isolation and characterization of a thermosensitive *OsSGS3a* mutant

To investigate the molecular mechanisms of rice stress responses, we identified a *thermosensitive abnormal palea (tsp)* mutant from mutant population generated by $^{60}$Co γ-radiation mutagenesis in an elite cultivar 2537. *tsp* developed open-glume florets and curved grains with significantly reduced yield traits in the summer of Shanghai with high field temperature (HT) frequently occurring at rice booting stage, but produced normal florets and grains in the winter season in Hainan with normal temperature (NT) suitable for rice growth (Fig. 1a, b, Supplementary Fig. 1a, b, and Supplementary Data 1). The pollen viability was not affected in *tsp* under high temperature (Supplementary Fig. 1c), thus the yield reduction was mainly caused by palea defects with disordered sclerenchyma (fs) and spongy parenchymatous cells (spc) in *tsp* (Supplementary Fig. 1d). These results suggest that the abnormal palea polarity identity likely leads to the floret defects in the mutant.

Genetic analysis revealed that the abnormal character of *tsp* was controlled by a single recessive locus. Further mapping combined with whole-genome sequencing identified only one annotated candidate gene, *Suppressor of Gene Silencing 3 (OsSGS3)*, which encodes a rice

homolog of *Arabidopsis* SGS3, with a single nucleotide deletion in the last exon of *OsSGS3a* in *tsp*. The single nucleotide deletion in *tsp* resulted in a substitution of valine (V568) to stop codon, leading to a truncated OsSGS3a protein (Fig. 1c, d and Supplementary Fig. 1e, f). The identity of *OsSGS3a* gene was further confirmed using genetic complementation (Fig. 1e and Supplementary Fig. 1 g, h). We therefore renamed this mutant as *ossgs3a-1*. Taken together, we conclude from these results that the *OsSGS3a* locus is responsible for the thermosensitive mutant phenotype.

### OsSGS3a interacts with OsSGS3b and functions as a major player in rice growth and development

We generated *OsSGS3a-Flag* transgenic plants in model rice variety Nipponbare (NIP) (Supplementary Fig. 2a), and performed immunoprecipitation-mass spectrometry (IP-MS) analysis to identify potential interacting partners of OsSGS3a. Interestingly, the homolog protein OsSGS3b (*LOC_Os12g09590*) was identified in the IP-MS sample from *OsSGS3a-Flag* transgenic plants, but not in the control NIP sample (Fig. 2a and Supplementary Data 2). Like AtSGS3, OsSGS3a and OsSGS3b both contain plant-specific domains (Supplementary Fig. 2b). We confirmed the interaction of OsSGS3a with OsSGS3b in yeast two-hybrid assay and found that the CC (coiled coil) domain of OsSGS3a was essential for their interaction (Fig. 2b). Split luciferase complementation (SLC) and co-immunoprecipitation (Co-IP) assays further confirmed the OsSGS3a/OsSGS3b interaction in *Nicotiana benthamiana* and rice (Fig. 2c, d). Like AtSGS3[38], OsSGS3a but not OsSGS3b can form a homodimer (Fig. 2c and Supplementary Fig. 2c), suggesting that the formation of such homodimer or heterodimer may play a role in plant siRNA biosynthesis.

*OsSGS3a* was highly expressed in young panicles and developing seeds, while *OsSGS3b* mainly in nodes and stems

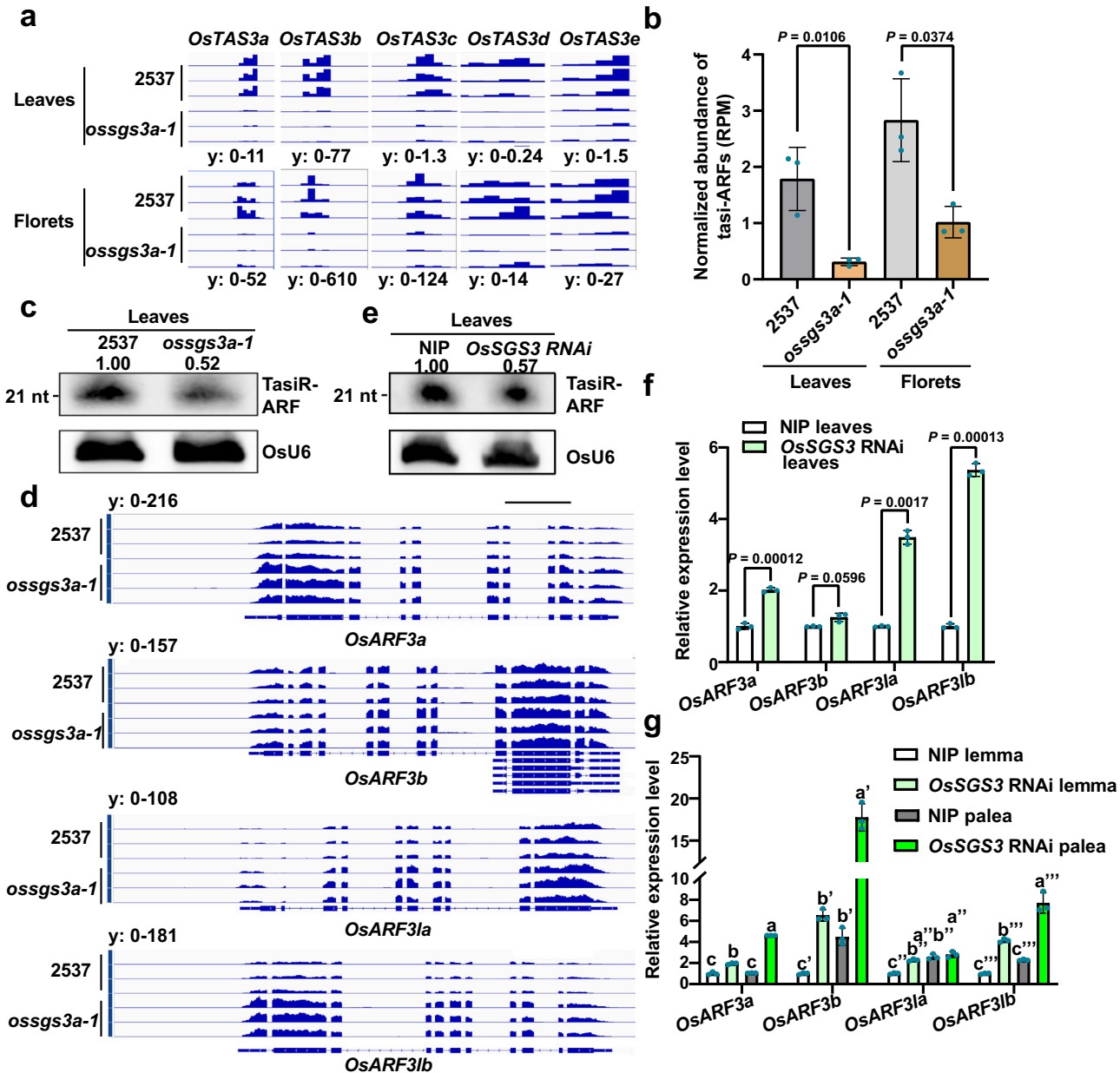

**Fig. 3 | OsSGS3a modulates the biogenesis of tasiR-ARF. a** Genome browser view of *OsTAS3a-e*-derived tasiRNAs in the leaves and florets of 2537 and *ossgs3a-1*. Leaves were sampled from 2-week-old seedlings grown in the growth chamber at 28 °C and florets were collected from 2537 and *ossgs3a-1* grown under high field temperature. **b** Normalized abundance of tasiR-ARFs in the leaves and florets of 2537 and *ossgs3a-1* as determined by small RNA deep sequencing. **c** RNA blot of tasiR-ARFs in the leaves of 2537 and *ossgs3a-1* grown in the field. **d** Genome browser view of the *OsARF3la, OsARF3lb, OsARF3a,* and *OsARF3b* mRNAs in the leaves of 2537 and *ossgs3a-1*. Scale bar, 1 kb. **e** The abundances of tasiR-ARFs in the leaves of NIP and *OsSGS3* RNAi transgenic plants grown in the field. *OsU6* was used as a loading control (**c, e**). The intensity of the blots was quantified (**c, e**). **f, g** Real-time RT-PCR analyses of relative expressions of *OsARF3la, OsARF3lb, OsARF3a,* and *OsARF3b* in the indicated samples grown in the field. *OsActin* served as an internal control (**f, g**). The average values (±s.d., *n* = 3, biologically independent samples) are shown (**b, f, g**). Significant differences were determined by two-tailed Student's *t* test (**b, f**) or one-way ANOVA with Tukey's HSD post hoc analysis (**g**). Exact *P* values are indicated above the bars (**b, f**) or provided in Supplementary Data 7 (**g**). Experiments were repeated three times with similar results (**c, e–g**). Source data are provided as a Source Data file (**b, c, e–g**).

(Supplementary Fig. 2d). This implies that *OsSGS3a* and *OsSGS3b* may function with subtle differences. To further investigate the function of OsSGS3a and OsSGS3b in rice, we tried to create *ossgs3a* and *ossgs3b* knockout transgenic plants in NIP background. Although at least three different types of mutations in *OsSGS3a* could be detected in the transformed calli (Supplementary Fig. 3a), no *ossgs3a* knockout transgenic plants were generated, suggesting that loss-of-function of *OsSGS3a* was most likely lethal. By contrast, we obtained *ossgs3b* knockout plants carrying different mutations (Supplementary Fig. 3b), which developed normal palea but decreased yield

values (Supplementary Fig. 3c–f). We further knockout *OsSGS3b* in *ossgs3a-1* background (Supplementary Fig. 4a, b) and found that all *ossgs3a-1ossgs3b* plants displayed irregular phyllotaxis and thread-like leaves and seedling lethality (Supplementary Fig. 4a). Moreover, we used RNA interference (RNAi) to simultaneously reduce the expression of *OsSGS3a* and *OsSGS3b* in transgenic plants (Supplementary Fig. 4c). The *ossgs3a/b* knockdown plants (*OsSGS3* RNAi) exhibited floret defects and sterility at high temperature (Supplementary Fig. 4d). Next, we also generated transgenic plants overexpressing *OsSGS3a/b-Flag*, which did not consistently change

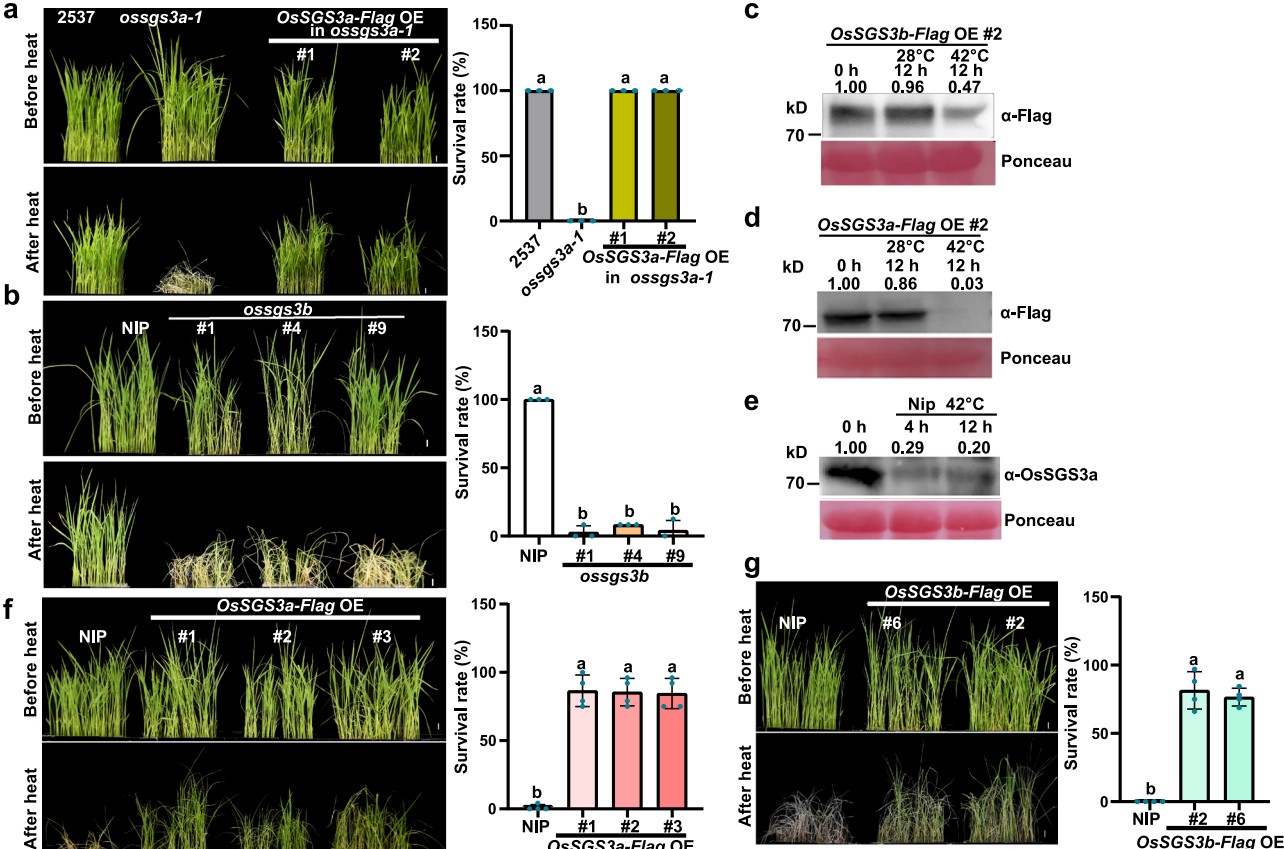

**Fig. 4 | OsSGS3a/b positively regulates thermotolerance in rice. a, b** Fewer *ossgs3a-1* (**a**) and *ossgs3b* (**b**) plants survived compared with respective wild-type plants after heat stress. In comparison with *ossgs3a-1*, the complemented transgenic lines displayed a similar survival rate as that of wild-type plants (**a**). 12-day-old seedlings were treated at 42 °C for 42 h (**a**) or 48 h (**b**) in a growth chamber and subsequently recovered for 2 days at 28 °C before being photographed. **c, d** The abundance of OsSGS3b-Flag (**c**) and OsSGS3a-Flag (**d**) protein decreased after heat stress treatment, as immunodetected with anti-Flag antibody. **e** OsSGS3a protein levels in the indicated samples. An anti-OsSGS3a antibody was used to detect endogenous OsSGS3a. Ponceau staining was used as a loading control (**c**–**e**). The intensity of the blots was quantified (**c**–**e**). Twelve-day-old seedlings were treated at 42 °C for the indicated hours in a growth chamber (**c**–**e**). **f, g** Overexpression of OsSGS3a-Flag or OsSGS3b-Flag can improve thermotolerance in rice. After heat stress treatment at 42 °C for 60 h in a growth chamber, seedlings were recovered at 28 °C for 2 days before being photographed. The survival rate was shown as mean ± s.d. (*n* = 3 or 4, biologically independent samples) (**a, b, f, g**). Scale bars, 1 cm (**a, b, f, g**). Significant differences were determined by one-way ANOVA with Tukey's HSD post hoc analysis (**a, b, f, g**). Exact *P* values are provided in Supplementary Data 7 (**a, b, f, g**). Source data are provided as a Source Data file (**a**–**g**). Experiments were repeated three times with similar results (**a**–**g**).

agronomic traits (Supplementary Figs. 2a and 5a–f), suggesting a feedback regulation. These results imply that OsSGS3a and OsSGS3b are partially functionally redundant with OsSGS3a acting as a major player in rice growth and development.

We next analyzed the subcellular localization of OsSGS3a and OsSGS3b and found that OsSGS3a and OsSGS3b colocalized in the cytoplasmic granules in rice protoplasts similar with siRNA bodies observed in *Arabidopsis*[39] (Supplementary Fig. 6a). Loss-of-function of OsSGS3a did not affect the localization of OsSGS3b and vice versa (Supplementary Fig. 6b).

## OsSGS3a plays an important role in tasiRNA production

A role for SGS3 proteins in the biogenesis of 21-nt secondary siRNAs has been proposed[28,40], and our previous studies have shown that AtSGS3 was involved in siRNA biogenesis in *Arabidopsis*[35,36]. We used high-throughput sequencing to examine small RNA populations in *ossgs3a-1* mutant and the wild-type 2537 grown at high temperature (Supplementary Data 3). The sRNA-seq analysis revealed a similar size distribution profile in 2537 and *ossgs3a-1* (Supplementary Fig. 7a, b). However, tasiRNAs derived from *OsTAS3a-e*[41,42] were hardly detected in the leaves and florets of *ossgs3a-1* (Fig. 3a), supporting its role in tasiRNA production in rice. As tasiR-ARFs have been shown to regulate rice development[43,44], we next used small RNA gel blots to confirm the accumulation of tasiR-ARFs. Consistently, tasiR-ARFs signals were significantly reduced in *ossgs3a-1* mutants but restored in complementation lines (Fig. 3b, c and Supplementary Fig. 7c, d).

Since tasiR-ARFs target *OsARF3la/OsARF14* (*LOC_Os05g43920*), *OsARF3lb* (*LOC_Os01g54990*), *OsARF3a/OsARF2* (*LOC_Os01g48060*), and *OsARF3b/OsARF15* (*LOC_Os05g48870*)[42,45], we observed that the decreased tasiR-ARFs indeed de-repressed the expression of *OsARF3s* in *ossgs3a-1* mutants (Fig. 3d). Similarly, in *OsSGS3* RNAi plants, the biogenesis of tasiR-ARFs was significantly compromised along with upregulation of *OsARF3s* (Fig. 3e, f and Supplementary Fig. 7e, f). The decreased accumulation of tasiR-ARFs resulted in the significant upregulation of *OsARF3s* in the palea (Fig. 3g). Loss-of-function of *OsSGS3b* or overexpression of *OsSGS3a/b-Flag* did not consistently affect the abundance of tasiR-ARFs or the expression of *OsARF3a/b/la/lb* in leaves (Supplementary Fig. 8a–f). Taken together, these results further support OsSGS3a as a critical regulator of tasiR-ARF biogenesis in rice. Therefore, we concluded that OsSGS3, tasiRNAs, and OsARF3s form a functional module in rice. Interestingly, a similar molecular module fine-tunes the lateral root growth in *Arabidopsis* through the feedback regulation of miR390 accumulation by AtARF2/3/4[46]. We also propose that the feedback regulatory mechanism may contribute to the unincreased accumulation of tasiR-ARFs in plants overexpressing *OsSGS3a/b*.

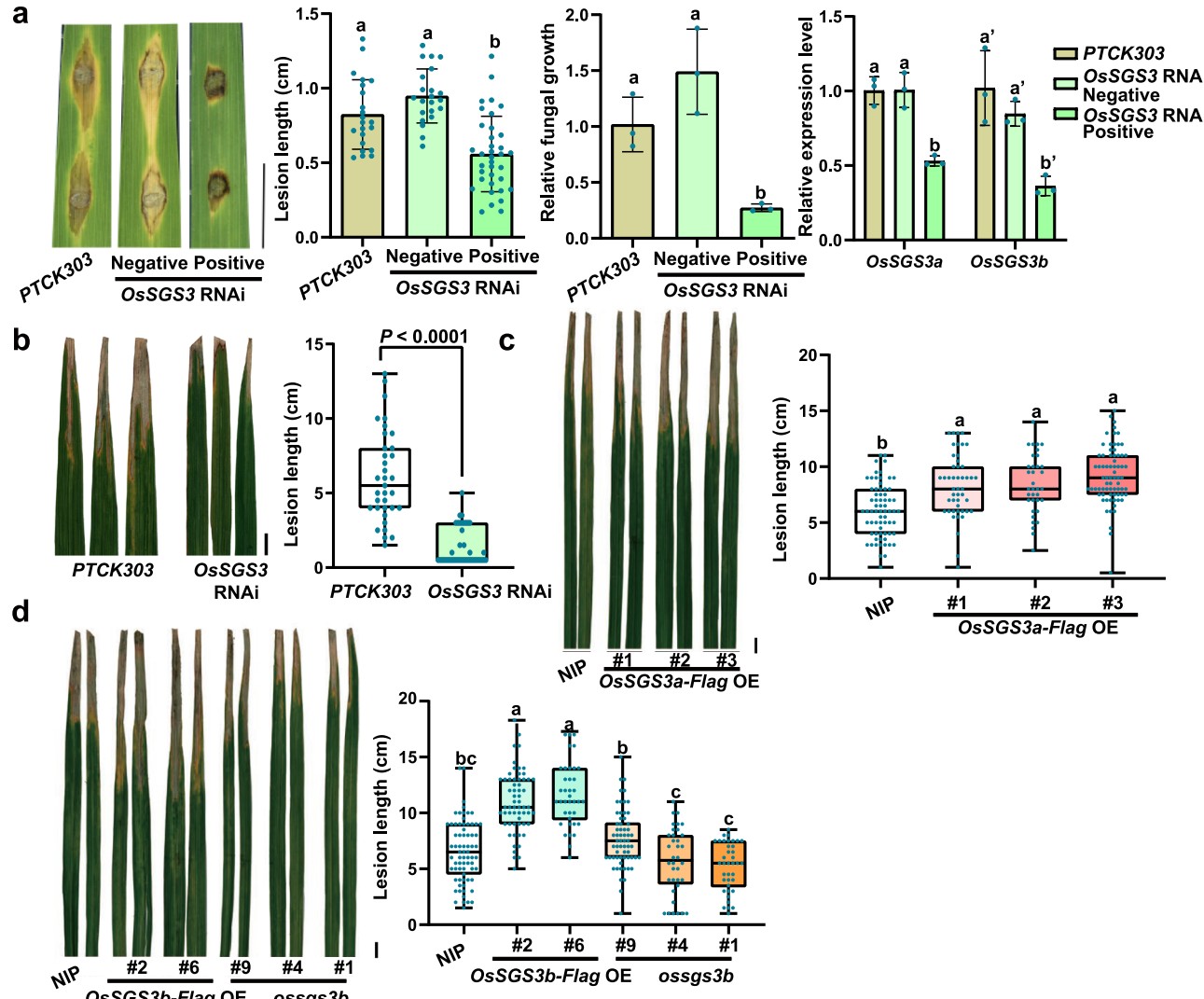

**Fig. 5 | OsSGS3a negatively regulates rice resistance to *Xoo* and *M. oryzae*.**
**a** Blast resistance of positive and negative *OsSGS3* RNAi lines at 7 dpi (days post inoculation) with punch injection inoculation (TH12). Left panel: Disease symptom of the indicated lines; Middle panels: lesion length (mean ± s.d.; *n* = number of biologically independent samples in the graph) were determined, and relative fungal growth (mean ± s.d.; *n* = 3, biologically independent samples) was determined with fungal *POT2* normalized to rice *Ubiquitin* by qRT-PCR. Right panel: levels of *OsSGS3a* and *OsSGS3b* was measured by qRT-PCR. Data are presented as means ± s.d. (*n* = 3, biologically independent samples). *OsActin* served as an internal control to normalize the expression of *OsSGS3a* and *OsSGS3b*. The expression levels of *OsSGS3a* and *OsSGS3b* are decreased in the positive transgenic *OsSGS3* RNAi lines, but not in the negative *OsSGS3* RNAi lines. The *PTCK303* empty vector transgenic plants were used as the control. **b**–**d** Bacteria blight resistance of positive *OsSGS3* RNAi (**b**), *OsSGS3a-Flag* OE (**c**), *OsSGS3b-Flag OE* and *ossgs3b* lines (**d**). Eight-week-old plant materials grown in field were inoculated with PXO99A (OD$_{600}$ = 1.0). Disease symptom and lesion lengths were measured at 14 dpi, shown as box plots (*n* > 34, biologically independent samples). Box plots show the median (central line) and interquartile range (IQR; from the 25th to 75th percentile); whiskers extend to minimum and maximum values within 1.5 times the IQR (**b**–**d**). Similar results were obtained from three independent experiments (**a**–**d**). Scale bars, 1 cm (**a**–**d**). Significant differences were determined by two-tailed Student's *t* test (**b**) or one-way ANOVA with Tukey's HSD post hoc analysis (**a**, **c**, **d**). Exact *P* values are indicated above the bars (**b**) or provided in Supplementary Data 7 (**a**, **c**, **d**). Source data are provided as a Source Data file (**a**–**d**).

## OsSGS3a/b positively regulates thermotolerance in rice

In the field, *ossgs3a-1* is sensitive to high temperatures at the reproductive stage. To determine that OsSGS3a modulates thermotolerance, we heat-stressed plants in a growth chamber and observed that fewer *ossgs3a-1* plants survived compared with wild-type plants when subjected to 42 °C treatment (Fig. 4a). Similarly, *ossgs3b* seedlings also displayed thermosensitive phenotypes (Fig. 4b), although they did not show developmental defects. We previously reported that AtSGS3 protein is subjected to heat-induced degradation in *Arabidopsis*[35]. To further examine the mechanism in rice, we found that the abundance of OsSGS3a-Flag and OsSGS3b-Flag proteins gradually decreased during heat stress (Fig. 4c, d). This heat-mediated OsSGS3a degradation was also supported using an OsSGS3a antibody to detect the endogenous OsSGS3a protein level (Fig. 4e). Interestingly, overexpression of OsSGS3a or OsSGS3b improved thermotolerance and the transgenic plants displayed significantly increased survival rate (Fig. 4f, g). Therefore, we concluded that OsSGS3a and OsSGS3b play a role in plant thermotolerance.

We propose that OsSGS3-mediated thermotolerance most likely involves tasiRNAs, given that *OsSGS3* RNAi plants, both *OsRDR6* RNAi and *OsDCL4* RNAi transgenic plants displayed decreased thermotolerance (Supplementary Fig. 9a, b). To determine the mechanism underlying OsSGS3a-mediated thermotolerance, we performed RNA sequencing and identified downregulated genes in *ossgs3a-1* under heat stress (Supplementary Fig. 9c and Supplementary Data 3 and 4). Clustering and gene ontology (GO) analysis showed that these genes are enriched for ROS-related pathways, such as antioxidant and peroxidase activity (Supplementary Fig. 9d), suggesting that

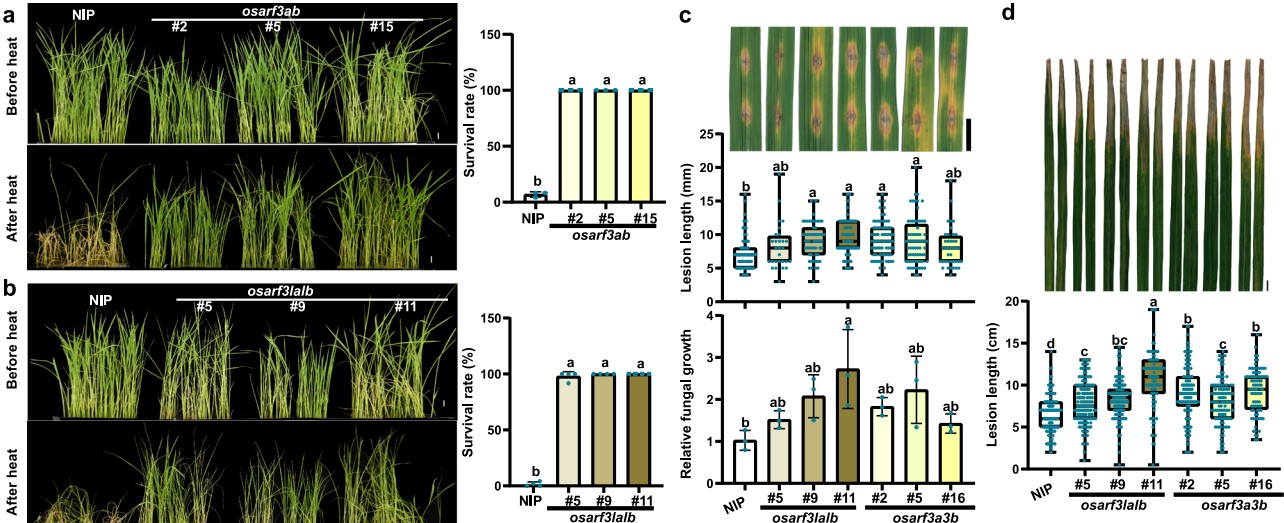

**Fig. 6 | OsARF3a/b/la/lb coordinates disease resistance and thermotolerance in rice. a, b** *osarf3ab* (**a**) and *osarf3lalb* (**b**) mutants displayed enhanced thermotolerance than the wild-type NIP. After heat stress treatment at 42 °C for 60 h in a growth chamber, seedlings were recovered at 28 °C for 2 days before photographed. The survival rate was shown as mean ± s.d. (*n* = 3 or 4, biologically independent samples). **c** Blast resistance of *osarf3lalb* and *osarf3a3b*. Disease symptom, lesion lengths and relative fungal growth were measured at 7 dpi. **d** Bacteria blight resistance of *osarf3lalb* and *osarf3a3b*. Disease symptom and lesion lengths were measured at 14 dpi. Lesion lengths were shown as box plots (*n* > 27, biologically independent samples) (**c, d**). Box plots show the median (central line) and interquartile range (IQR; from the 25th to 75th percentile); whiskers extend to minimum and maximum values within 1.5 times the IQR (**c, d**). Scale bars, 1 cm (**a–d**). Significant differences were determined by one-way ANOVA with Tukey's HSD post hoc analysis (**a–d**). Exact *P* values are provided in Supplementary Data 7 (**a–d**). Source data are provided as a Source Data file (**a–d**). All experiments were repeated three times with similar results.

OsSGS3a may modulate rice thermotolerance through ROS scavenging. Consistent with the compromised expression of ROS-related genes, DAB staining revealed increased accumulation of $H_2O_2$ in heat-stressed *ossgs3a-1* and *ossgs3b* mutant plants, *OsSGS3* RNAi, *OsRDR6* RNAi, and *OsDCL4* RNAi transgenic plants, which was associated with cell damage (Supplementary Fig. 9e–g). Similar thermo-sensitivity-associated ROS accumulation and cell death were also observed in our previous report[47]. Taken together, these results suggest the role of OsSGS3a/b in protecting rice cells from heat damage.

### OsSGS3a negatively regulates disease resistance to bacterial and fungal pathogens

Plants often prioritize defense over growth to defend against invading pathogens, which is known as the growth-defense trade-off[11,48]. We previously reported that decreased accumulation of AtSGS3 greatly compromised immunity in *Arabidopsis*[35]. As the bacterial leaf blight caused by *Xoo* and fungal blast caused by *M. oryzae* are two of the most devastating diseases in rice, we tested plant resistance to the two pathogens. To our surprise, in contrast with *Arabidopsis*, we found that *OsSGS3* RNAi plants displayed enhanced resistance to *M. oryzae* (Fig. 5a and Supplementary Fig. 10a), whereas *ossgs3b* plants showed no obvious change in blast resistance (Supplementary Fig. 10b, c). We did not observe consistently increased susceptibility in *OsSGS3a-Flag* or *OsSGS3b-Flag* plants (Supplementary Fig. 10b–e), probably due to the similar proposed feedback regulation. OsSGS3a protein level was slightly induced upon *M. oryzae* inoculation and slightly decreased later (Supplementary Fig. 10f). We further detected enhanced resistance to *Xoo* in *OsSGS3* RNAi plants, and overexpression of *OsSGS3a* or *OsSGS3b* significantly increased disease severity as in comparison with wild-type plants (Fig. 5b–d). By contrast, *ossgs3b* plants displayed no disease changes after inoculation with *Xoo* (Fig. 5d), suggesting that OsSGS3a may also play a major role in rice immunity. We propose that OsSGS3-mediated immunity most likely through tasiRNAs, given that *OsRDR6* RNAi and *OsDCL4* RNAi transgenic plants both displayed enhanced resistance to *Xoo* (Supplementary Fig. 10g, h).

Taken together, the results indicated that OsSGS3a plays a negative role in disease resistance in rice.

### OsARF3a/3b/la/lb coordinates disease resistance and thermotolerance in rice, probably through ROS homeostasis

To determine tasiRNAs and their downstream targets that modulate antagonistic thermotolerance and defense in rice, we used small RNA gel blotting to further confirm the heat-induced suppression of the biogenesis of tasiR-ARFs in NIP (Supplementary Fig. 11a). RNA profiling revealed that expression levels of *OsARF3a* and *OsARF3la* were up-regulated in both heat-stressed wild-type and *ossgs3a-1* mutants (Supplementary Fig. 11b). These data suggested that *OsARF3s* may modulate plant thermotolerance.

*OsARF3la*, *OsARF3lb*, *OsARF3a*, and *OsARF3b* all displayed high expression in rice leaves, and *OsARF3b* and *OsARF3lb* were also highly expressed in young panicles and developing seeds (Supplementary Fig. 12a). OsARF3la, OsARF3lb, OsARF3a, and OsARF3b were mainly localized in the nucleus in rice protoplasts (Supplementary Fig. 12b), consistent with their predicted functions as transcription factors. To elucidate the role of OsARF3s in plant thermotolerance and immunity, we used CRISPR/Cas9 technology to generate *osarf3ab* and *osarf3lalb* knockout mutants (Supplementary Fig. 13a–d). Both *osarf3ab* and *osarf3lalb* mutants displayed enhanced thermotolerance than wild-type NIP (Fig. 6a, b). When compared with NIP, less accumulation of $H_2O_2$ was observed in these mutants after heat treatment (Supplementary Fig. 13e). Taken together, these results indicated that OsARF3a/b/la/lb negatively regulates thermotolerance in rice at the seedling stage.

We next checked the possible roles of *OsARF3s* in plant immunity. We used *osarf3ab* and *osarf3lalb* homozygous mutants for disease resistance evaluation. Both *osarf3ab* and *osarf3lalb* homozygous mutants exhibited increased susceptibility to *M. oryzae* and *Xoo* (Fig. 6c, d). It has been reported that tasiR-ARFs were rapidly induced by *Xoo* at 2 h post infection (hpi), along with down-regulation of *OsARF3a and OsARF3b*, but no induction was observed at 24 hpi[49]. Intriguingly, we found that the accumulation levels of tasiR-ARFs were

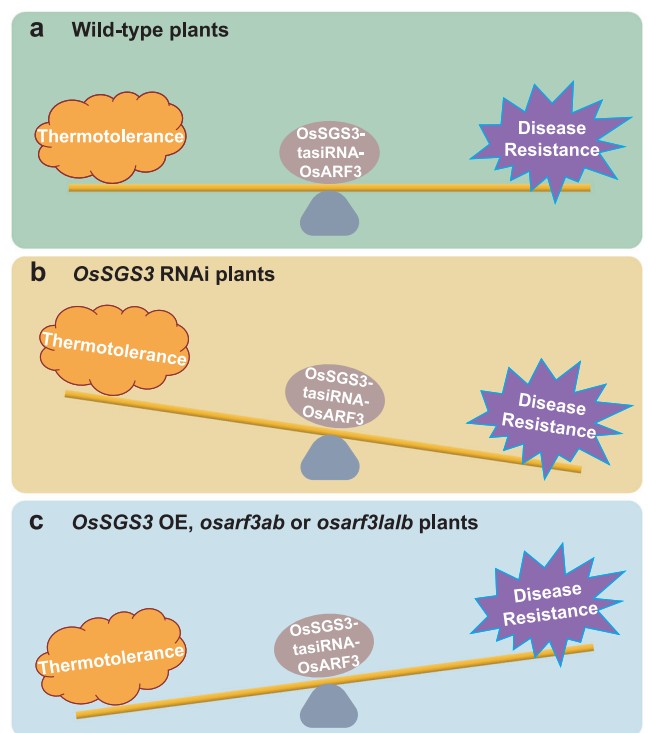

**Fig. 7 | A proposed model for OsSGS3-tasiRNA-OsARF module-mediated abiotic-biotic stress response trade-off. a** In wild-type rice, the OsSGS3-tasiRNA-OsARF3 module plays an important role in coordinating the trade-off between thermotolerance and disease resistance, which positively regulates thermotolerance, but negatively modulates immunity in rice. **b** In *OsSGS3* RNAi plants, disease resistance is enhanced to defend against invading pathogens at the cost of thermotolerance. **c** In *OsSGS3* OE, *osarf3ab* or *osarf3lalb*, plants prioritize thermotolerance over disease resistance.

also slightly increased in *OsSGS3* RNAi plants upon *M. oryzae* infection (Supplementary Fig. 13f, g and Supplementary Data 5), suggesting that unknown additional regulator(s) of tasiR-ARF production may respond to pathogen infection. Taken together, these results demonstrate that OsARF3a/b/la/lb coordinates disease resistance and thermotolerance in rice.

To determine the mechanism underlying OsARF3a/b/la/lb-mediated thermotolerance, we performed RNA sequencing and identified differentially expressed genes in wild-type NIP, *osarf3ab* and *osarf3lalb* knockout mutants under heat stress (Supplementary Data 3 and 4). Among the ROS-related genes that are modulated by OsSGS3a (Supplementary Fig. 9c), we identified a ROS scavenging-related gene *OsCATA* encoded by *LOC_Os02g02400*, which accumulated to higher levels in heat-stressed *osarf3ab* and *osarf3lalb* knockout mutants in comparison with wild-type (Supplementary Fig. 14a, b). These results suggest a potential role of OsARF3a/b/la/lb in repressing the expression of *OsCATA*.

Similar to *osarf3ab* and *osarf3lalb* knockout mutants, heat-stressed *osarf3a* seedlings[50] displayed thermo-tolerant phenotypes and less accumulation of $H_2O_2$ (Supplementary Fig. 14c, d), suggesting that OsARF3a negatively regulates rice thermotolerance at seedling stage. ARF proteins have been reported to modulate the expression of target genes through binding the auxin-responsive elements (AuxREs: TGTCNN)[51,52] or sugar-responsive elements (SuREs: GTCTC)[50]. To confirm the direct binding of OsARF3s to the promoter region of *OsCATA*, we obtained *OsARF3a-Flag* transgenic plants in *osarf3a*[50] and performed chromatin immunoprecipitation (ChIP)-qPCR. The result showed that OsARF3a-Flag could directly bind to the P4 fragment in the promoter of *OsCATA* that contained 1 AuxRE and 2 SuREs, but not to other fragments under heat stress (Supplementary Fig. 14e). The

regulatory role of OsARF3a on *OsCATA* expression and how OsCATA coordinates disease resistance and thermotolerance in rice will be further investigated independently.

## Discussion

For stable and sustainable food production in modern agriculture that is facing changing or even deteriorative environments, breeding crops with disease resistance and abiotic stress tolerance, two major targets of the multi-resistance strategy, provides a practical solution. However, because of the antagonism between abiotic stress responses and biotic (pathogen) defense, this multi-resistance breeding strategy has been a large challenge to plant breeders. Few studies have been conducted on defense-abiotic tolerance, resulting in a lack of knowledge on the trade-offs between disease resistance and abiotic tolerance in plants. Here we show that OsSGS3-tasiRNA-OsARF3 module plays an important role in coordinating the trade-off between heat tolerance and disease resistance, which positively regulates thermotolerance, but negatively modulates immunity in rice (Fig. 7). Interestingly, the same module likely modulates contrary outputs of the abiotic-biotic stress response trade-off in *Arabidopsis*[35], suggesting that different plants (rice as a summer plant vs. *Arabidopsis* as a winter plant) may adopt different trade-offs of abiotic-biotic stress responses.

ARFs can activate or repress the expression of their downstream targets, such as GH3 family members, which catalyze the formation of conjugated forms of indole-3-acetic acid (IAA) to maintain hormonal homeostasis[52]. In *Arabidopsis*, GH3 member WES1-mediated auxin homeostasis fine-tunes growth, abiotic and biotic stresses adaption responses[53]. OsARF12 positively modulates the expression of *OsGH3-2*, which contributes to rice basal resistance by suppressing pathogen-induced IAA accumulation[54–56]. Whether the OsSGS3-tasiRNA-OsARF3 module also recruits these GH3 targets to modulate thermotolerance and immunity will be further investigated. Our study suggested that OsARF3a may modulate the expression of *OsCATA*, which functions in scavenging ROS and rice immunity[57]. How OsCATA controls ROS homeostasis and coordinates disease resistance and thermotolerance remains to be studied.

The *miR390-TAS3-ARF* pathway is an evolutionarily conserved regulatory circuit in land plants[28,58]. In rice, the expression of OsmiR390 is downregulated by various abiotic and biotic stresses, such as heavy metal stress, drought and salt stress, UV light stress, and fungal infection[59]. Here, our results revealed that loss-of-function of *OsARF3a/b* and *OsARF3la/lb* lead to enhanced thermotolerance with slightly but statistically significantly reduced disease resistance, suggesting that tasiR-ARF and the OsARF3 targets may constitute an autoregulatory network to fine-tune rice immunity and thermotolerance. Notable, this fine-tuning of stress responses is mostly subtle without complete loss of disease resistance or thermotolerance, allowing plants to survive the diverse abiotic and biotic stresses.

Loss-of-function of AtSGS3 accelerates the juvenile-to-adult phase transition with elongated and epinastic leaves in *Arabidopsis*[40,60]. The mutation of *LEAFBLADELESS1* (*LBL1*), an ortholog of *AtSGS3*, leads to abaxialized radial leaves and abnormal reproductive development in maize[61]. Our study reveals that OsSGS3a is required for both vegetative and reproductive development in rice. SGS3 proteins most likely play important roles in abiotic and biotic stress responses through its function in the biogenesis of tasiRNAs. Interestingly, heat stress promotes AtSGS3[35] and OsSGS3a/b degradation, indicating that SGS3 is a key thermosensitive component. This heat-sensitive degradation of SGS3s may have a fundamentally biologically importance: redirecting resources for seed development thus ensuring environmental fitness under stress[35]. However, the abundance of OsSGS3a protein was slightly increased upon *M. oryzae* infection in wild-type plants, suggesting a differential modulation of OsSGS3 protein levels under biotic and heat stress. Importantly, we have showed that OsSGS3a and OsSGS3b enhanced rice thermotolerance while the suppression of

OsSGS3a/b enhanced rice disease resistance. Therefore, the OsSGS3-tasiRNA-OsARF3 module is likely differentially regulated in face of abiotic and biotic stress. Possible natural alleles or gene editing of SGS3 genes may provide valuable resources for breeding thermotolerant as well as disease-resistant crops.

In paddy field, rice plants are often exposed to the combination of different abiotic stresses and pathogens. OsSGS3-tasiRNA-OsARF3 module positively regulates abiotic stress response, but negatively modulates plant disease resistance. This regulatory module of multiple components may have an evolutionary advantage: allowing the network to preciously respond to diverse stresses under the agricultural conditions. More studies with abiotic and biotic stress mutants will provide a comprehensive insight into the genetic interactions between the two important stress response machineries in plants. Nevertheless, our current study provides an experimental example that the trade-off between thermotolerance and plant immunity can be fine-tuned by the OsSGS3-tasiRNA-OsARF3 module in crop breeding.

## Methods

### Plant materials and growth conditions

The *japonica/Geng* variety Nipponbare (NIP) was used to generate various transgenic plants, including *OsSGS3a-FLAG* OE, *OsSGS3b-FLAG* OE, *ossgs3b*, *OsSGS3 RNAi*, *OsRDR6* RNAi, *OsDCL4* RNAi, *osarf3ab*, and *osarf3lalb*. The *tsp/ossgs3a-1* mutant was isolated from the cultivar 2537 and used to generate *OsSGS3a-Flag OE* in *ossgs3a-1* and *ossgs3a-1ossgs3b*. All rice plants were grown in the isolated experimental fields in Shanghai (East China, summer rice season), or Hainan (South China, winter rice season) under natural conditions for seed production, pathogen inoculation, and phenotypic analysis. Plants were grown in Shanghai in the summer with high field temperature (HT, ≥35 °C) frequently occurring at the rice booting stage, or in Hainan in the winter with normal temperature (NT) suitable for rice growth (Supplementary Data 1). For thermotolerance analysis of plants at the seedling stage, 12-day-old seedlings were treated at 42 °C in growth chambers for the indicated periods shown in Figs. 4a, b, f, g, 6a, b, Supplementary Figs. 9b and 14c. Seed-setting rate, 1000-grain weight, and full-filled grain yield per plant were quantified during field trials. *Nicotiana benthamiana* for transient expression was grown at 22 °C with a 16 h/8 h photoperiod.

### Histochemical analyses

The accumulation of hydrogen peroxide ($H_2O_2$) was detected by 3,3′-diaminobenzidine (DAB) staining as described previously[62]. Heat-stressed rice leaves were collected and vacuum-infiltrated in DAB solution [1 mg/mL DAB, 10 mM MES, pH 3.8, 0.2% (v/v) Tween-20] for 5 min and incubated at 25 °C for 8 h in the light. Samples were then cleared by boiling in 96% ethanol for 10 min before photography. Semi-thin sectioning experiments were performed as previously described[45]. In brief, samples were fixed in FAA (50% ethanol, 5% glacial acetic acid, and 5% formaldehyde) at 4 °C overnight and then dehydrated in an ethanol series. Samples were then embedded in Technovit-7100 for sectioning. The 4 µm thick sections were cut with a microtome (HM340E), stained with toluidine blue, and observed under a microspore.

### Map-based cloning

Mapping populations were generated by crossing *tsp* mutants with IRTA129. Using defective glume plants identified in the $F_2$ segregating population, the locus was first mapped to chromosome 12 within a 2.41 M interval. Further, the *tsp* was backcrossed to 2537 for whole-genome sequencing using the Illumina NovaSeq6000 platform, which was performed by Berry Genomics. The SNP index for mutant at segregating bi-allelic SNPs in 100-kb sliding windows (by 10 kb) was summed for the various sequencing pools, and allele frequencies were calculated. Finally, the difference in allele frequency (SNP index) between the mutant pool and wild-type pool was calculated for all pairwise comparisons and plotted across the 12 chromosomes.

### Constructs and rice transformation

For the construction of *OsSGS3a* or *OsSGS3b* overexpression plasmid, full-length coding sequence of *OsSGS3a* or *OsSGS3b* was amplified from genomic DNA, fused with Flag tag or not, and cloned into binary vector *PUN1301* under the control of the ubiquitin promoter (*Ubi*) of *Zea mays*. To construct the *OsSGS3*, *OsRDR6*, and *OsDCL4* knockdown plasmid, 300-bp cDNA fragments of *OsSGS3*, *OsRDR6*, and *OsDCL4* were amplified, respectively, and inserted into the binary vector *pTCK303* in both the sense and antisense orientations. Driven by ubiquitin promoter, RNAs were transcribed from the *pTCK303-OsSGS3*, *pTCK303-OsRDR6*, and *pTCK303-OsDCL4* vectors, were then cleaved by Dicer proteins to generate siRNAs to guide the gene silencing of endogenous *OsSGS3*, *OsRDR6*, and *OsDCL4* transcripts, respectively. For CRISPR/Cas9 constructs, the designed sgRNAs of *OsSGS3a*, *OsSGS3b*, *OsARF3a*, *OsARF3b*, *OsARF3la*, and *OsARF3lb* were synthesized and fused into the *pYLCRISPR* vector following the protocol[63]. All the constructs were introduced into *Agrobacterium tumefaciens* strain EHA105 to transform NIP calli via Agrobacterium-mediated transformation, leading to the generation of more than 30 independent lines for each transgene. PCR-based sequencing was performed to identify target gene mutations and homozygous mutants were used for experiments. All primer sequences used for cloning can be found in Supplementary Data 6.

### Pathogen inoculation and disease resistance assay

*Xoo* and blast inoculation were performed as previously described[57,62]. For the *Xoo* resistance assay, 2-month-old rice seedlings grown in the field were inoculated with Philippine strain P6 (PXO99A) suspended in sterilized water at a concentration of $OD_{600} = 1.0$ by leaf-clipping method[64]. Upon *Xoo* infection, lesions usually develop as water-soaked to yellow-orange stripes on mechanically injured parts of leaves, and progress toward the leaf base. Lesion lengths were measured from more than 30 leaves for each genotype and recorded at 14 dpi (days post inoculation). *M. oryzae* isolate TH12 was used for rice blast inoculation. For seedling spraying inoculation with *M. oryzae* spores ($1 \times 10^5$ spores/mL in sterile water containing 0.05% Tween-20), two-week-old seedlings were grown in a dew growth chamber at 26 °C with a 14 h/10 h (day/night) photoperiod. For punch inoculation, leaves of 4-week-old rice seedlings grown in the field were punch inoculated with spore suspensions ($1 \times 10^5$ spores/mL in sterile water containing 0.05% Tween-20). At 5−7 dpi, lesion lengths of fifteen infected leaves were measured. Blast lesions are spindle-shaped with necrotic borders. The relative fungal growth was calculated by DNA-based quantitative PCR (qPCR) using the threshold cycle value ($C_T$) of *M. oryzae Pot2* DNA against the $C_T$ of rice genomic *UBIQUITIN* DNA. All experiments were independently repeated at least two times.

### RNA analysis

Total RNAs were extracted using TRIzol reagent (Invitrogen) from rice leaves or florets. To prevent loss of small RNAs with low GC content, $MgCl_2$ was added at a final concentration of 0.2 M[65]. RNA was quantified using NanoDrop, and ~1 µg total RNA was converted into cDNA using HiScript Q RT SuperMix with gDNA remover (Vazyme, R123-01). cDNAs were amplified for quantitative analysis using SYBR Premix Ex Taq (Takara, RR420A) following the manufacturer's instructions using a CFX96 Touch Real-Time PCR Detection System (Bio-Rad). *OsActin* served as an internal control to normalize the expression of individual genes. The primers used for qRT-PCR are listed in Supplementary Data 6.

To analyze small RNAs, 20−30 µg total RNA were separated by electrophoresis on 15% polyacrylamide/7 M urea gels, transferred to nylon membrane (GE Health), and subjected to UV irradiation-mediated cross-linking. Biotin-labeled probes complementary to

specific siRNA were synthesized and hybridized with the membrane. The signals of immobilized nucleic acids were detected by Nucleic Acid Detection Module Kit (Thermo Fisher, 89880). After stripping the probes, OsU6 probe was used to re-probe the same membrane as a loading control. Uncropped versions of all gels or blots are shown in the Source Data files. qRT-PCR analysis of tasiR-ARF was conducted as previously described[43]. *OsU6* served as an internal control to normalize the expression of tasiR-ARF. The primers are listed in Supplementary Data 6.

## Protoplast transient expression assay

The fusion protein expression plasmids pA7-OsSGS3a-eGFP, pA7-GFP-OsSGS3a, pA7-OsSGS3b-eGFP, pA7-GFP-OsSGS3b, pA7-OsSGS3a-mCherry, pA7-OsSGS3b-mCherry, pA7-OsARF3a-eGFP, pA7-OsARF3b-eGFP, pA7-OsARF3la-eGFP, and pA7-OsARF3lb-eGFP were constructed and purified with HiSpeed Plasmid Midi Kit (QIAGEN, 12643). The NLS sequence was fused with RFP into PA7-35S-RFP as a nuclear marker. CFP driven by the 35 S promoter was used as a non-specific localized protein control. The isolation of rice protoplasts from seedling sheaths and PEG-mediated transformation were performed as described previously[66]. Fluorescence signals were detected using a confocal microscope (Olympus Fluoview FV1000 and Leica TSC SP8 STED 3X).

## Protein analysis

HIS-OsSGS3a-N (amino acids 1–184) was used as an antigen to produce anti-OsSGS3a polyclonal antibodies by ABclonal® Technology (Wuhan, China). To extract total proteins from plants, 0.2 g fresh leaves were ground into fine power in liquid nitrogen and homogenized in the protein extraction buffer (150 mM Tris-HCl, pH 7.5, 6 M urea, 2% SDS, and 5% β-mercaptoethanol). Extracts were boiled for 5 min and then centrifuged at 13,000 rpm for 10 min at 4 °C to remove debris. Supernatants were collected and mixed with SDS loading buffer for protein gel blot analysis.

The co-immunoprecipitation (Co-IP) was performed as described previously[66]. Protein extracts from rice leaves were prepared in the IP lysis buffer (50 mM Tris-HCl, pH 7.5, 150 mM NaCl, 1 mM EDTA, 10% glycerol, 1% Triton X-100, 1 mM PMSF, and 1× protease inhibitor cocktail). Different combinations of supernatants were incubated with pre-washed Anti-FLAG® M2 Magnetic Beads (Sigma, M8823) at 4 °C for 2 h and washed four times with lysis buffer. The bound proteins were eluted from the affinity beads by boiling for 5 min in 40 µl 2× SDS loading buffer and analyzed by immunoblotting. Antibodies, including anti-GFP (Abcam, ab290, 1:2000), anti-FLAG (Sigma, F1804, 1:2000), anti-OsSGS3a (custom-developed by ABclonal® Technology, China, 1:1000), anti-ACTIN (CMCTAG, AT0004, 1:2000), Goat Anti-Rabbit IgG HRP (Thermo, 31460, 1:10,000), Goat Anti-Mouse IgG HRP (BIO-RAD, 170-6516, 1:10,000), were used. Co-IP and western blot experiments were independently repeated at least two times. Uncropped versions of all gels or blots are shown in the Source Data files.

## LC-MS/MS analysis

To identify OsSGS3a-interacting proteins, total proteins were extracted from the leaves of 2-week-old seedlings of *OsSGS3a-FLAG* OE with the IP lysis buffer mentioned above. Protein extracted from NIP seedlings was used as a negative control. The OsSGS3a-FLAG proteins were enriched by immunoprecipitation using Anti-FLAG® M2 Magnetic Beads, washed four times with IP lysis buffer, eluted with 0.1 M glycine (pH = 2.5). The supernatant was collected and separated by SDS-PAGE, and protein bands were digested into peptides by trypsin and recovered. Peptides mixtures were analyzed by Easy-nLC 1200 (Thermo Scientific) and mass spectrometer (Orbitrap Exploris 480, Thermo Scientific). The data were analyzed with MaxQuant software (1.6.17.0) for protein identification and quantification, using the Rice Annotation Project (RAP) database (http://rapdb.dna.affrc.go.jp/). The results of LC-MS/MS analysis are provided in Supplementary Data 2.

## Yeast two-hybrid assay

The coding sequences of OsSGS3a and OsSGS3b as well as different versions of truncated OsSGS3a proteins were cloned into pDEST22 (AD) or pDEST32 (BD) vectors. The desired combinations of bait and prey constructs were co-transformed into yeast strain AH109. Clones were incubated on selective mediums lacking Trp, Leu, Ade and His. Images were taken 3 days after incubation.

## SLC experiments

For SLC, OsSGS3a, OsSGS3aΔCC, and OsSGS3b were fused to the N- or C-terminus of firefly luciferase and transformed to *Agrobacterium* strain GV3101. Overnight cultures of *Agrobacteria* were collected by centrifugation and resuspended in MES buffer (10 mM MgCl$_2$,10 mM MES, 150 µM acetosyringone, pH 5.6). The desired combinations of clones were mixed with GV3101 expressing P19 at a final OD$_{600}$ of 0.5 and incubated at 28–30 °C for 2–3 h before infiltration. Leaves of 4-week-old *N. benthamiana* were syringe-infiltrated with the suspensions. After 2 days, the LUC activity was measured with Luciferase Assay Systems (Promega), and images were captured using the Tanon-5200 Chemiluminescent imaging system (Tanon). For luciferase activity measurement, infiltrated leaves were collected and ground into fine powder in liquid nitrogen, and homogenized in lysis buffer following the manufacturer's instructions (Promega). The supernatant was then incubated with luciferin substrate in a 96-well plate for 10–15 min, and luminescence was measured with the Varioskan Flash (Thermo Fisher Scientific) plate reader. All primers used are listed in Supplementary Data 6.

## mRNA sequencing and data analysis

Libraries for mRNA-seq were constructed and sequenced by Annoroad Gene Technology (Beijing). Briefly, RNAs were quantified by NanoDrop 2000c UV-Vis Spectrophotometer (Thermo Fisher Scientific), agarose gel electrophoresis, and Agilent 2100 Bioanalyzer (Agilent technologies). mRNAs were isolated from total RNAs by poly(A) selection, fragmented into short fragments, and converted to cDNAs. cDNAs were ligated to adapters and the suitable fragments were selected for PCR amplification as templates. All the RNA-seq libraries were pair-end sequenced on an MGI DNBSEQ T7 platform.

mRNA sequencing data analysis were performed as reported[65]. Low-quality reads were removed, and adapters were trimmed to obtain clean reads, which were mapped to the reference genome (MSU v7). The expression level of each gene was calculated as the FPKM value (fragments per kilobase of transcript per million mapped reads). For differential gene expression analysis, we used a ≥1.5-fold change and a false discovery rate (FDR) ≤0.05 as cutoffs.

## Small RNA sequencing and data analysis

Small RNA libraries construction and data analysis were performed as reported[65]. Total RNAs enriched for small RNA were extracted from heat-stressed or normal leaves and young florets of *ossgs3a-1* mutant plants and the wild-type controls. Total RNAs were ligated to pre-adenylated 3′ adaptors by T4 RNA ligase 2, truncated KQ (NEB, M0373L), and then ligated to 5′ RNA adaptors using T4 RNA ligase 1 (NEB, M0204L). These RNAs were reverse transcribed and amplified using GXL DNA polymerase (Takara, R050A). PCR products between 135 and 160 bp in length were collected by gel separation and pair-end sequenced on an Illumina HiSeqX-Ten platform by Annoroad Gene Technology (Beijing).

For small RNA-sequencing data analysis, low-quality reads were filtered and 3′ adapter sequences were trimmed. Clean small RNA reads between 18 to 30 nt in length were mapped to the rice reference genome (MSU v7) using the program Bowtie, allowing no mismatches. Small RNA reads that mapped to the rRNAs, tRNAs, snRNAs, and snoRNAs were filtered.

## ChIP-qPCR analysis

ChIP experiments were performed as described[67]. Briefly, 1 g two-week-old seedling leaves of *OsARF3a-Flag*/*osarf3a*[50] and ZH11 (Negative control) were harvested after heat treatment at 42 °C for 6 h and cross-linked with 1% formaldehyde. After quenching the cross-linking by adding glycine, the leaves were ground into fine powder in liquid nitrogen. Chromatin was isolated and sheared into 200 to 500-bp fragments by sonication. The sonicated chromatin was incubated with antibodies against Flag (Sigma, F1804) overnight at 4 °C. Immuno-complexes were precipitated, washed, eluted, and the cross-linking was reversed overnight at 65 °C. Precipitated DNA fragments were then recovered with the GlycoBlue™ Coprecipitant (Thermo, AM9516) according to the manufacturer's instructions. ChIP-qPCR was performed with three replicates, and relative fold enrichments were indicated by the percentage of input DNA. ChIP experiments were performed independently twice with similar results. Sequences of the primers used for ChIP-qPCR are listed in Supplementary Data 6.

## Statistical analysis

For qRT-PCR analyses, thermotolerance assays, agronomic traits assessment, and disease assays, significant differences between samples/lines and the corresponding controls were analyzed using two-tailed Student's *t* test for pairwise comparisons, one-way or two-way ANOVA analysis with Tukey's multiple comparison test as specified in the Figure legends. Samples sharing lowercase letters are not significantly different. Detailed information about statistical analysis values for all ANOVA analyses is provided in Supplementary Data 7.

## Reporting summary

Further information on research design is available in the Nature Portfolio Reporting Summary linked to this article.

## Data availability

All data are available within this article and its Supplementary Information. The mRNA and small RNA-sequencing datasets generated in this study have been deposited in the Genome Sequence Archive in National Genomics Data Center (NGDC) under project CRA008524 (https://ngdc.cncb.ac.cn/gsa/browse/CRA008524). The mass spectrometry proteomics data have been deposited to the ProteomeXchange Consortium via the PRIDE[68] partner repository with the dataset identifier PXD043706. A reporting summary for this article is available as a Supplementary Information file. Original data points in graphs and uncropped versions of all gels or blots are shown in the Source Data files. Source data are provided with this paper.

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

## Acknowledgements

We thank Professor Yijun Qi (Tsinghua University) and Pengfei Jiang (Anhui Agricultural University) for kindly providing protocols for constructing small RNA libraries. We also thank Professor Xiang Yu (Shanghai Jiao Tong University) for assistance in analyzing the data of small RNA sequencing. We thank Dr. Leonard Krall for critical reading and comments. We thank the proteomics facility of Life Sciences School in Yunnan University for LC-MS/MS analysis. This work was supported by funding from the National Natural Science Foundation of China (91940301 and 32088102 to Z.H., 32070564 and 31600207 to J.Z.L., 32170620 to X.S.), the Chinese Academy of Sciences (XDB27040201 to Z.H., XDA24010302 to X.S.), and Yunnan Fundamental Research Projects (202101AW070002 and 202201AT070090 to J.Z.L.).

## Author contributions

X.G., F.S., X.S., Z.H., and J.Z.L. conceived and designed the experiments; X.G., F.S., and S.L. performed most of the experiments, including thermotolerance assay, histochemical analyses, construction of plasmids

and rice transformation, pathogen inoculation and disease resistance assay, RNA analysis, protoplast transient expression assay, protein analysis, yeast two-hybrid assay and SLC assay; Z.F., C.Y., and Y.Y. performed small RNA libraries construction and analyzed the RNA-sequencing data; D.L. performed pathogen inoculation and disease resistance assay, RNA analysis, protoplast transient expression assay and protein analysis in the revision; D.L., P.Y., J.T., T.L., L.L., J.L.Z., J.Z.L., and L.F. performed agronomic traits analysis; B.Y. and Y.Z.Y. performed map-based cloning; P.Y. and X.N.W. performed LC-MS/MS analysis; J.L. performed yeast two-hybrid assay; X.G., F.S., L.F., J.Y.L., Y.Z.Y., Z.Z., J.W., Y.D., X.S., and J.Z.L. developed materials; X.N.W., Y.D., and X.C. provided theoretical contributions to the project; X.G., F.S., Z.F., S.L., X.S., Z.H., and J.Z.L. analyzed the data and wrote the paper. All authors read and approved the final manuscript.

## Competing interests

The authors declare no competing interests.
