## [Peer Review File · Nature Communications]

The OsSGS3-tasiRNA-OsARF3 module orchestrates abiotic-biotic stress response trade-off in riceREVIEWER COMMENTS

Reviewer #1 (Remarks to the Author):

In this study, the authors identified the OsSGS3a, which positively regulates thermotolerance, but modulates contrary immunity in rice. Furthermore, OsARF3 family members, the targets of OsSGS3-dependent tasiRNAs, are also included in the abiotic-biotic stress response. This is an interesting manuscript and the experiments are well-performed except for some comments that are discussed below. Overall, clarification is needed for figure legends, and some figures are small.

Major comments

1) The function of OsSGS3a-FLAG OE

The authors performed a complementation analysis of *ossGS3a-1* using OsSGS3a OE (Fig.1e and SupFig1.g). Does the “OsSGS3a OE” means the “OsSGS3a-FLAG OE”? If these lines are different, did you perform the complementation analysis, OsSGS3a-FLAG OE in *ossGS3a-1*? I would like to know the effect of Tag fused OsSGS3a in Figure 4.

2) OsSGS3 protein level in disease resistance

How about the OsSGS3 proteins after blast or bacteria treatment in WT? The authors have an anti-OsSGS3 antibody, which was used in Fig4. e. Please show the OsSGS3 protein level in blast or bacteria resistance (*Xoo* and *M.oryzae*) in WT using the western-blotting analysis.

Mainor comments

1) Please explain “CK” in fig.4 c and d.

2) What are the “positive and negative” in Fig.5a? It is difficult for readers, which are not familiar with pathogens.

3) The figures of SupFig1.d are small, so it is difficult to detect the defect of *fs* and *spc* in mutants.

Reviewer #2 (Remarks to the Author):

This is an intriguing study that demonstrates a role for the rice SGS3a protein in the trade-off between thermotolerance and pathogen defense response in this important grain crop. This is a timely and interesting study that identifies a somewhat different role for the SGS3a protein in these stress responses in a monocot crop species as compared to the genetic model dicot *Arabidopsis*. Thus, this study provides important new findings in the field of small RNA-mediated regulation of plant phenotypes. I also find it intriguing that rice has two homologs of SGS3 that seem to have only semi-redundant functions, but also appear to have specific functionalities and more careful characterization of their redundant and unique functionalities will be intriguing to follow. This study is for the most part well done and supports the major conclusions well. However, there are a couple of missing experiments in my opinion that I outline below. There are also necessary manuscript edits that are needed in a revised manuscript version before this study is ready for publication in this journal.

Necessary Experimental Additions

1. OsRDR6 and OsDCL4 need to be tested for their functions in thermotolerance using the RNAi lines that this group has in their position. They measured the effects of these proteins on pathogen resistance so they should also be studied for their effects on thermotolerance.

2. I also believe that to be published in a high impact journal at least some of the genes involved in ROS production should be confirmed as direct binding targets of the OsARF3s. This can be done by ChIP-qPCR of selected targets or by ChIP-seq of one or more of the OsARF3s. At the very least an RNA-seq in their new osarf mutants demonstrating the opposite levels of these transcripts in the absence of these proteins as compared to the ossgs3a mutant would be nice evidence of this direct link that can then be probed further in future studies.

Necessary Editorial Revisions:

1. The Introduction is a bit long and could be streamlined especially the discussion on the TFs known to be involved in abiotic and biotic stress responses

2. I am not sure what “functions as a major player” means in the second Results subsection title. This needs to be edited to be more informative and addresses what the results reveal about the function of OsSGS3a.

3. The manuscript is quite well written but there are some grammatical errors throughout (misuse of the, missing the, etc.) that need to be corrected in a revised version of the manuscript.

Reviewer #3 (Remarks to the Author):

This manuscript details investigation into rice genes involved in abiotic and biotic stress responses. The authors identify OsSGS3a which was shown to be involved in modulating rice thermotolerance as an ossga3a mutant was unable to tolerate high-temperature conditions. Additionally, a relationship was established between OsSGS3a and the synthesis of tasiR-ARFs. OsARFa/b were shown to be involved in positively regulating disease resistance. This manuscript provides novel insight in the ways OsARF3a/b are involved in promoting rice resistance to bacterial blight and rice blast disease while OsSGS3a negatively regulates disease resistance. Moreover, these results reinforce studies that show disease resistance is compromised during abiotic stresses. This paper describes the relationship between abiotic stresses, like heat stress, and defense against bacterial and fungal pathogens. In all, this paper describes in depth the trade-off of rice plants in prioritizing disease resistance over general growth. There is an importance in studying the relationship between thermotolerance and immunity as it brings new insight into the factors affecting rice crops and may provide an avenue to explore for mitigating stresses in the field.

This manuscript is very well-written and organized. The authors chose and conducted experiments that were aligned with the questions asked. The introduction incorporates the background of how heat is impactful to rice growth and development for a general audience. The introduction also promotes the relationships between heat stresses and RNA production. The discussion includes the importance of these findings to crop production and management, a concept that is validated within the manuscript. The overall findings of this manuscript are novel and bring new insight into the field of rice stress responses and pathogen immunity. Additionally, these findings could potentially be translated to other plant systems.

Major comments:

During heat stress tests, the authors are not clear on the exact temperatures that constitute normal field temperatures and high field temperatures. The authors should clarify these specific conditions during the initial time they mention these experiments.

Additionally, the authors should clarify the growth conditions for the rice. Is the rice grown in a greenhouse or crop fields? How are the authors accounting for the variability of temperatures?

While figure 1 d depicts the shift of band for the truncated OsSGS3a protein from the WT, the authors should include the molecular weight of both proteins to validate western blot results.

Supplementary Figure 1 g shows the overexpression of OsSGS3a was done in the mutant strain *oss3a-1* background. This figure should include the expression results from the mutant background only as a control.

Supplemental figure 6 should have a control protein. The authors should have either GFP by itself or an unrelated protein tagged with a fluorescent protein to show the change in localization when OsSGS3 is tagged with GFP.

In supplementary figure 6b, there is a different localization based on the terminal where the fluorescent protein is located. It is not consistent with the stated claims that there is no deviation in localization. Tagging different regions of the proteins have an impact on localization, and a different localization pattern is seen for both OsSGS3a and OsSGS3b.

Supplementary figure 8 needs statistical significance.

Supplementary figure 8 a-c shows variety in the different transgenic lines but #2 and #6 depict a big variation and the quantification numbers 0.75 and 1.05 don't seem to match with the variability. This could signify that the transgenic lines are not very uniform. In lines 226-229, the authors state there is no variability in the ARF overexpression, but there is variability in the expression of *tasiR-ARFs*.

Figure 5: the labeling of positive and negative RNAi lines is confusing. What this means here. This figure needs a regular control like the WT to depict normal infection.

The supplementary figure 10 b infection profile is very variable. Were all spore concentrations equal throughout infections and was the same method for infection used? Some infections look like punch inoculations and others look like spray inoculation. The authors need to have consistent infections across all lines.

The authors state that reducing the expression of OsSGS3 promotes more resistance to pathogens. However, figure 5 b shows less bacterial blight in the OsSGS3 RNAi lines, but there is evident lesions present in the rice, consistent with fungal infection. What is this due to? Is this presenting bacterial blight infection or *M. oryzae* infection? This figure seems to show cross-contamination.

This manuscript should provide a description of how bacterial and fungal infection looks like in rice. The infection phenotypes need to be established for readers to know how typical infections present in rice, especially when working with different phytopathogens.

Minor edits:

Figure legends should be consistent throughout all figures. Abbreviations should be included in the legend. For example, figure 1 should include what NT and HT mean.

Figure 4: what is CK? Abbreviations should be included in the figure legend.

Line 64: suppress to suppresses

Line 279: wild type misspelled

We appreciate the constructive comments made by the reviewers. We have provided additional data and revised our manuscript to address the concerns raised by the reviewers. Please find our detailed point-by-point response to reviewer comments below.

Reviewer #1 (Remarks to the Author):

In this study, the authors identified the OsSGS3a, which positively regulates thermotolerance, but modulates contrary immunity in rice. Furthermore, OsARF3 family members, the targets of OsSGS3-dependent tasiRNAs, are also included in the abiotic-biotic stress response. This is an interesting manuscript and the experiments are well-performed except for some comments that are discussed below. Overall, clarification is needed for figure legends, and some figures are small.

Re: We thank the reviewer for the very positive evaluation of our study and the constructive comments on the manuscript. We have taken into account all your suggestions and thoroughly revised the text.

Major comments

1) The function of OsSGS3a-FLAG OE

The authors performed a complementation analysis of *oss3a-1* using OsSGS3a OE (Fig.1e and SupFig1.g). Does the “OsSGS3a OE” means the “OsSGS3a-FLAG OE”? If these lines are different, did you perform the complementation analysis, OsSGS3a-FLAG OE in *oss3a-1*? I would like to know the effect of Tag fused OsSGS3a in Figure 4.

Re: Thanks for the comments. The “*OsSGS3a* OE in *oss3a-1*” in Fig. 1e and Supplemental Fig. 1g-h means the “*OsSGS3a-Flag* OE in *oss3a-1* background”. We have renamed these lines in the revised version. The complementation lines generated non-truncated OsSGS3 protein and developed normal plants as the wild-type (Fig. 1e and Supplemental Fig. 1g-h). Therefore, we propose that OsSGS3a-Flag is functionally equivalent as OsSGS3 in the complementation assay.

To further determine the effect of OsSGS3a-Flag in *oss3a-1* background, we have

analyzed the accumulation of tasiR-ARFs and found that the expression of *OsSGS3a-Flag* restored the production of tasiR-ARFs in the complementation lines (Supplemental Fig. 7d). Moreover, in comparison with *oss3a-1*, the transgenic complementation plants displayed a significantly increased survival rate and decreased accumulation of H₂O₂ upon heat stress (Fig. 4a and Supplemental Fig. 9e). Taken together, our results confirm that *OsSGS3a* locus is responsible for the mutant phenotype.

For the thermotolerance assay in Fig. 4a, we used the *oss3a-1* seeds that were generated from high temperature-grown plants (termed as HT *oss3a-1* seeds in Fig. 1a) in our previous version. However, HT *oss3a-1* generated few seeds (Fig. 1b), which were not enough for new experiments. In the revised version, we used the *oss3a-1* seeds harvested from normal temperature-grown plants (termed as NT *oss3a-1* seeds in Fig. 1a), for the thermotolerance assay. Similarly, fewer NT *oss3a-1* seedlings survived compared with wild-type plants after heat stress. We now used the new data as Fig. 4a.

2) OsSGS3 protein level in disease resistance

How about the OsSGS3 proteins after blast or bacteria treatment in WT? The authors have an anti-OsSGS3 antibody, which was used in Fig4. e. Please show the OsSGS3 protein level in blast or bacteria resistance (*Xoo* and *M.oryzae*) in WT using the western-blotting analysis.

Re: We thank the reviewer for these very insightful suggestions. We have analyzed the protein abundance of endogenous OsSGS3a in the leaves of 2-week-old wild-type NIP before and after inoculation with *M. oryzae*. The OsSGS3a protein level was slightly induced at 12 h and 24 h post inoculation (hpi) and slightly decreased at the later timepoints (Supplemental Fig. 10f in the new version of manuscript). As the levels of tasiR-ARFs were not significantly increased in NIP plants at 24 hpi upon *M. oryzae* infection (Supplemental Fig. 13g), the slightly increased accumulation of OsSGS3a protein at 24 hpi may not significantly affect the production of tasiR-ARFs.

We also checked the OsSGS3a protein level in the leaves of 5-week-old wild-type NIP before and after inoculation with *Xoo*. However, due to the extremely low

expression of *OsSGS3a* in the leaves at the growth stage (Supplemental Fig. 2d), we can hardly detect the signals of OsSGS3a protein (Author Response Figure 1, for review only). We proposed that the abundance of OsSGS3a protein may decrease in leaves when rice grows, probably age-dependent, given that OsSGS3a was easily detected in 2-week-old seedling leaves but not in 5-week-old plant leaves. However, it is far beyond the goal of our current study.

Author Response Figure 1 (For review only). Immunoblot analysis of OsSGS3a protein abundance during a time course of 0~72 h in the leaves of wild-type inoculated with *Xoo*.

5-week-old plants were infected with *Xoo* strain PXO99A, and protein was extracted from infected and water mock control leaves collected at different infection time points. OsActin served as a loading control.

Minor comments

1) Please explain “CK” in fig.4 c and d.

Re: CK in Fig. 4c and 4d indicates the corresponding seedlings grown at 28°C. We have

changed it to 28°C.

2) What are the “positive and negative” in Fig.5a? It is difficult for readers, which are not familiar with pathogens.

Re: Thank you for raising this point. The “positive” indicates the *OsSGS3* RNAi seedlings in which the expression levels of *OsSGS3a* and *OsSGS3b* are decreased when compared with the control *PTCK303* transgenic plants (positive RNAi plants), which carry the empty vector. The “negative” indicates the *OsSGS3* RNAi plants in which the expression levels of *OsSGS3a* and *OsSGS3b* are similar as that of the control (negative RNAi plants). In the revised figure, we added the *PTCK303* empty vector transgenic plants as the control. In comparison with the *PTCK303* and negative *OsSGS3* RNAi plants, the positive *OsSGS3* RNAi plants displayed enhanced resistance to *M. oryzae* (Fig. 5a).

3) The figures of SupFig1.d are small, so it is difficult to detect the defect of fs and spc in mutants.

Re: Sorry for this. We have enlarged the figure in Supplemental Fig. 1d and indicated silicified cells (sc), fibrous sclerenchyma (fs), spongy parenchymatous cells (spc), nonsilicified cells (nc). In the enlarged figure, the disordered fs and spc in *tsp* are clearly presented.

Reviewer #2 (Remarks to the Author):

This is an intriguing study that demonstrates a role for the rice SGS3a protein in the trade-off between thermotolerance and pathogen defense response in this important grain crop. This is a timely and interesting study that identifies a somewhat different role for the SGS3a protein in these stress responses in a monocot crop species as compared to the genetic model dicot Arabidopsis. Thus, this study provides important new findings in the field of small RNA-mediated regulation of plant phenotypes. I also find it intriguing that rice has two homologs of SGS3 that seem to have only semi-redundant functions, but also appear to have specific functionalities and more careful characterization of their redundant and unique functionalities will be intriguing to follow. This study is for the most part well done and supports the major conclusions well. However, there are a couple of missing experiments in my opinion that I outline below. There are also necessary manuscript edits that are needed in a revised manuscript version before this study is ready for publication in this journal.

Re: We thank the reviewer for the thorough summary of our work and the constructive comments on the manuscript.

Necessary Experimental Additions

1. OsRDR6 and OsDCL4 need to be tested for their functions in thermotolerance using the RNAi lines that this group has in their position. They measured the effects of these proteins on pathogen resistance so they should also be studied for their effects on thermotolerance.

Re: We thank the reviewer for this constructive suggestion. We measured the performance of *OsSGS3* RNAi, *OsRDR6* RNAi, and *OsDCL4* RNAi transgenic plants under heat stress, and used the wild-type NIP and *PTCK303* empty vector transgenic plants as controls. Similar to the thermo-sensitive *ossGS3a-1*, fewer *OsSGS3* RNAi, *OsRDR6* RNAi, and *OsDCL4* RNAi plants survived compared with the controls when subjected to 42°C treatment (Supplemental Fig. 9b). DAB staining revealed increased accumulation of H₂O₂ in heat-stressed *OsSGS3* RNAi, *OsRDR6* RNAi, and *OsDCL4* RNAi plants (Supplemental Fig. 9g). These results suggested that the OsSGS3-

mediated thermotolerance most likely involves tasiRNAs.

2. I also believe that to be published in a high impact journal at least some of the genes involved in ROS production should be confirmed as direct binding targets of the OsARF3s. This can be done by ChIP-qPCR of selected targets or by ChIP-seq of one or more of the OsARF3s. At the very least an RNA-seq in their new *osarf* mutants demonstrating the opposite levels of these transcripts in the absence of these proteins as compared to the *ossGS3a* mutant would be nice evidence of this direct link that can then be probed further in future studies.

Re: We thank the reviewer for this insightful comment. To determine the mechanism underlying OsARF3a/3b/1a/1b-mediated thermotolerance, we performed RNA sequencing and identified differentially expressed genes in the wild-type NIP, *osarf3ab*, and *osarf3lalb* knockout mutant plants under heat stress (Supplemental Data 2 and 3 in the new version). Among the reactive oxygen species (ROS)-related genes that are modulated by OsSGS3a (Supplemental Fig. 9c in the new version), we identified a ROS scavenging-related gene *OsCATA* encoded by LOC_Os02g02400, which accumulated to higher levels in heat-stressed *osarf3ab* and *osarf3lalb* knockout mutant plants compared with wild-type (Supplemental Fig. 14a and 14b). These results suggest a role of OsARF3a/3b/1a/1b in repressing the expression of *OsCATA*.

Similar to *osarf3ab* and *osarf3lalb* knock-out mutant plants, heat-stressed *osarf3a* seedlings (Zhao *et al.*, 2022) displayed thermo-tolerant phenotypes and less accumulation of H₂O₂ (Supplemental Fig. 14c and 14d), suggesting that OsARF3a negatively regulates thermotolerance in rice at the seedling stage. ARF proteins have been reported to modulate the expression of target genes through binding the auxin-responsive elements (AuxREs: TGTCNN) (Guilfoyle & Hagen, 2007; Chapman & Estelle, 2009; Chandler, 2016; Cancé, *et al.*, 2022) or sugar-responsive elements (SuREs: GTCTC) (Zhao *et al.*, 2022). There are 18 AuxREs and 5 SuREs in the promoter of *OsCATA* (Supplemental Fig. 14e). To confirm the direct binding of OsARF3s to the promoter region of *OsCATA*, we obtained *OsARF3a-Flag* transgenic plants in the *osarf3a* background (Zhao *et al.*, 2022) and performed chromatin

immunoprecipitation (ChIP)-qPCR. Given that the germination rate of *OsARF3a-Flag/osarf3a* was very low, we did not have enough seedlings to perform ChIP assay to test the binding of OsARF3a to *OsCATA* promoter under normal temperature, heat stress, and pathogen infection. We only treated *OsARF3a-Flag/osarf3a* and ZH11 seedlings at 42°C for 6 h and performed ChIP-qPCR assays. The results revealed that under heat stress, OsARF3a could directly bind to the P4 fragment in the promoter of *OsCATA* that contained 1 AuxRE and 2 SuREs, but not to other fragments (Supplemental Fig. 14e). We have added these results to the revised manuscript (Supplemental Fig. 14). The regulatory role of OsARF3a on *OsCATA* expression need to be further biochemically and genetically determined.

We previously reported a Ca²⁺ sensor mutant *rod1*, which showed autoimmunity phenotypes with cell death and H₂O₂ accumulation (Gao *et al.*, 2021). ROD1 recruits and promotes catalase activity. In another independent investigation, we observed that over-expression of *OsCATA* in *rod1* reduced plant resistance against *Xoo* as in comparison with *rod1* (Author Response Figure 2a, for review only). Moreover, the transgenic plants exhibited an increased thermotolerance (Author Response Figure 2b, c, for review only). These results suggest that *OsCATA* coordinates disease resistance and thermotolerance in rice. As we are studying on H₂O₂-mediated heat responses and immunity as an independent story, we show these results here for review only.

Author Response Figure 2 (For review only). OsCATA may coordinate disease resistance and thermotolerance in rice.

a *OsCATA* OE in *rod1* seedlings displayed increased susceptibility to *Xoo* than *rod1*. Disease symptom and lesion lengths were measured at 14 dpi. Levels of *OsCATA* RNA was measured by qRT-PCR. Data are presented as means \pm s.d. ($n = 3$). *OsActin* served as an internal control to normalize the expression of *OsCATA*.

b Overexpression of *OsCATA* can improve thermotolerance in rice. After heat stress treatment at 42°C for 45 hours, seedlings were recovered at 28°C for 2 days before being photographed. The survival rate was shown as mean \pm s.d. ($n = 3$, biologically independent samples). Significant differences were determined by two-tailed Student's *t*-test for pairwise comparisons (**a**, **b**).

c DAB staining revealed decreased accumulation of H₂O₂ in heat-stressed *OsCATA* OE in *rod1* plants. 12-day-old seedlings were treated at 42°C for 24 h or 45 h in a growth chamber before DAB staining. Scale bars, 1 cm (**a**, **b**, **c**).

References:

- Cancé, C., Martin-Arevalillo, R., Boubekour, K. & Dumas, R. Auxin response factors are keys to the many auxin doors. *New Phytol.* **235**, 402-419 (2022).
- Chandler, J.W. Auxin response factors. *Plant Cell Environ.* **39**, 1014-1028 (2016).
- Chapman, E.J. & Estelle, M. Mechanism of auxin-regulated gene expression in plants. *Annu. Rev. Genet.* **43**, 265-285 (2009).
- Gao, M. *et al.* Ca²⁺ sensor-mediated ROS scavenging suppresses rice immunity and is exploited by a fungal effector. *Cell* **184**, 5391-5404.e5317 (2021).
- Guilfoyle, T.J. & Hagen, G. Auxin response factors. *Curr. Opin. Plant Biol.* **10**, 453-460 (2007).
- Zhao, Z. *et al.* Auxin regulates source-sink carbohydrate partitioning and reproductive organ development in rice. *Proc. Natl. Acad. Sci. U. S. A.* **119**, e2121671119 (2022).

Necessary Editorial Revisions:

1. The Introduction is a bit long and could be streamlined especially the discussion on the TFs known to be involved in abiotic and biotic stress responses

Re: Thank you for this good suggestion. We have streamlined the introduction in the revised manuscript. We also shortened the introduction part by modifying sentences

2. I am not sure what “functions as a major player” means in the second Results subsection title. This needs to be edited to be more informative and addresses what the results reveal about the function of OsSGS3a.

Re: Many thanks for pointing this out. Following your comment, we have changed the title to “OsSGS3a interacts with OsSGS3b and functions as a major player in rice growth and development”.

3. The manuscript is quite well written but there are some grammatical errors throughout (misuse of the, missing the, etc.) that need to be corrected in a revised version of the manuscript.

Re: Thanks for this comment. We have carefully checked the grammatical errors throughout the manuscript and corrected them in the revised version.

Reviewer #3 (Remarks to the Author):

This manuscript details investigation into rice genes involved in abiotic and biotic stress responses. The authors identify OsSGS3a which was shown to be involved in modulating rice thermotolerance as an *ossga3a* mutant was unable to tolerate high-temperature conditions. Additionally, a relationship was established between OsSGS3a and the synthesis of tasiR-ARFs. OsARFa/b were shown to be involved in positively regulating disease resistance. This manuscript provides novel insight in the ways OsARF3a/b are involved in promoting rice resistance to bacterial blight and rice blast disease while OsSGS3a negatively regulates disease resistance. Moreover, these results reinforce studies that show disease resistance is compromised during abiotic stresses. This paper describes the relationship between abiotic stresses, like heat stress, and defense against bacterial and fungal pathogens. In all, this paper describes in depth the trade-off of rice plants in prioritizing disease resistance over general growth. There is an importance in studying the relationship between thermotolerance and immunity as it brings new insight into the factors affecting rice crops and may provide an avenue to explore for mitigating stresses in the field.

This manuscript is very well-written and organized. The authors chose and conducted experiments that were aligned with the questions asked. The introduction incorporates the background of how heat is impactful to rice growth and development for a general audience. The introduction also promotes the relationships between heat stresses and RNA production. The discussion includes the importance of these findings to crop production and management, a concept that is validated within the manuscript. The overall findings of this manuscript are novel and bring new insight into the field of rice stress responses and pathogen immunity. Additionally, these findings could potentially be translated to other plant systems.

Re: We thank the reviewer for the appreciation of our work.

Major comments:

During heat stress tests, the authors are not clear on the exact temperatures that constitute normal field temperatures and high field temperatures. The authors should clarify these specific conditions during the initial time they mention these experiments.

Re: Many thanks for pointing this out. Usually, rice suffers high temperature ($\geq 35^{\circ}\text{C}$) and prefers normal temperature ($20\text{-}28^{\circ}\text{C}$). We usually grow rice during June-Oct in Shanghai, where the daily high temperature of the summer often surpassed 35°C (Shen

et al. 2015).

tsp developed open-glume florets and curved grains under high field temperature (HT) with significantly reduced yield traits in Shanghai in the summer, whereas produced normal florets and grains in Hainan Island in the winter rice season (Normal field temperature, NT) (Fig. 1a, b, Supplemental Fig. 1a, b). We have collected the daily temperatures in Shanghai and Hainan from the local weather stations (Supplemental Data 1). We have added the daily highest temperatures in each figure legend and methods in the revised version of manuscript.

Reference:

Shen, H. *et al.* Overexpression of receptor-like kinase ERECTA improves thermotolerance in rice and tomato. *Nature Biotechnology* **33**, 996-1003 (2015).

Additionally, the authors should clarify the growth conditions for the rice. Is the rice grown in a greenhouse or crop fields? How are the authors accounting for the variability of temperatures?

Re: For phenotypic assessments of 2537 and *tsp/oss3a-1* in Fig. 1 and Supplemental Fig. 1, plants were grown in Shanghai in the summer with high field temperature (HT) frequently occurring at the rice booting stage, or in Hainan in winter with normal temperature (NT) suitable for rice growth. We have collected the daily temperature in Shanghai and Hainan from the local weather stations from 2019~2021 (Supplemental Data 1). In August and early September at the Shanghai station at rice booting stage, the daily highest temperature was 36°C in 2019, 37°C in 2020, and 36°C in 2021 (Supplemental Data 1). While in February at the Hainan station, the daily highest temperature was 30°C in 2020~2022, which is suitable for rice growth (Supplemental Data 1).

For thermotolerance analysis of plants at the seedling stage, 12-day-old seedlings were treated at 42°C in growth chambers for the indicated periods shown in Fig. 4a, b, f, g, and 6a, b, Supplemental Fig. 9b, and 14c. Seed-setting rate, 1000-grain weight, and full-filled grain yield per plant were quantified during field trials as indicated in the

figure legend. For the *Xoo* resistance assay, 2-month-old rice seedlings grown in the field were inoculated with Philippine strain P6 (PXO99A). For seedling spraying inoculation with *M. oryzae* spores, two-week-old seedlings were grown in a dew growth chamber at 26°C with a 14 h/10 h (day/night) photoperiod. For punch inoculation, leaves of 4-week-old rice seedling grown in the field were punch inoculated with spore suspensions. We have described the growth conditions of rice in each figure legend in the revised manuscript.

While figure 1 d depicts the shift of band for the truncated OsSGS3a protein from the WT, the authors should include the molecular weight of both proteins to validate western blot results.

Re: Thanks for pointing this out. We have added the sentence “OsSGS3 protein (69.4 kD) was detected in 2537 while the truncated OsSGS3a protein (64.6 kD) was produced in *tsp*” in the figure legend of Fig. 1.

Supplementary Figure 1 g shows the overexpression of OsSGS3a was done in the mutant strain *oss3a-1* background. This figure should include the expression results from the mutant background only as a control.

Re: Thanks for the comments. In *oss3a-1*, the single nucleotide deletion resulted in a substitution of valine (V568) to stop codon, leading to a truncated OsSGS3a protein. In the RT-PCR assay, the primers of OsSGS3a cannot distinguish the transcripts of *OsSGS3a* in 2537 and *oss3a-1*. Instead, we analyzed the expression of OsSGS3a protein in the florets of 2537, *oss3a-1*, and two complementation lines grown under high field temperature. It is needed to note that the “*OsSGS3a* OE in *oss3a-1*” in Fig. 1e and Supplemental Fig. 1g-h means the “*OsSGS3a-Flag* OE in *oss3a-1*”. We have renamed these lines in the revised version. In addition to the truncated OsSGS3a protein, OsSGS3a-Flag protein can be detected in the complementation lines. The band of OsSGS3a-Flag protein was slightly higher than the band of OsSGS3a protein in 2537 in the gel. The complementation lines generated non-truncated OsSGS3 protein and developed normal floret as the wild-type (Fig. 1e and Supplemental Fig. 1g-h). To

further determine the effect of OsSGS3a-Flag in *oss3a-1* background, we have analyzed the accumulation of tasiR-ARFs and found that the expression of *OsSGS3a-Flag* restored the production of tasiR-ARFs in the complementation lines (Supplemental Fig. 7d). Moreover, in comparison with *oss3a-1*, the transgenic complementation plants displayed significantly increased survival rate and decreased accumulation of H₂O₂ (Fig. 4a and Supplemental Fig. 9e). Taken together, our results confirm that *OsSGS3a* locus is responsible for the mutant phenotype.

Supplemental figure 6 should have a control protein. The authors should have either GFP by itself or an unrelated protein tagged with a fluorescent protein to show the change in localization when OsSGS3 is tagged with GFP.

Re: Many thanks for the good suggestions. We have transformed OsSGS3a-eGFP and mCherry or OsSGS3b-eGFP and mCherry in NIP protoplasts. Florescence signals were detected using a confocal microscope (Leica TSC SP8 STED 3X). The signals of mCherry were higher in comparison with the GFP signals. We provided plotting of pixel intensities for the different color channels along transects. Note that the gray values of signals obtained from Leica confocal microscope were lower than that of Olympus Fluoview FV1000, which may be caused by the differences between the two machines. Although weakly expressed, some OsSGS3a-eGFP or OsSGS3b-eGFP proteins still displayed localization in the cytoplasmic granules (Supplemental Fig. 6a). Co-expression of OsSGS3a and OsSGS3b proteins may promote the formation of OsSGS3-containing cytoplasmic granules.

In supplementary figure 6b, there is a different localization based on the terminal where the fluorescent protein is located. It is not consistent with the stated claims that there is no deviation in localization. Tagging different regions of the proteins have an impact on localization, and a different localization pattern is seen for both OsSGS3a and OsSGS3b.

Re: We agreed with the reviewer that N-terminal or C-terminal tagged fluorescent protein might result in different localization pattern of both OsSGS3a and OsSGS3b

protein, especially OsSGS3a protein in *oss3b #1*. Our repeated experiments displayed similar results (Author Response Figure 3, for review only). However, in the plotting of pixel intensities for the different color channels along transects, the signals of N-terminal GFP-tagged or C-terminal mCherry-tagged OsSGS3a/OsSGS3b always merged with the signals of CFP (Supplemental Fig. 6b), which served as a non-specific localized marker. These results suggested that loss-of-function of OsSGS3a did not affect the co-localization of OsSGS3b with non-specific localized CFP and vice versa.

repeat1

repeat2

Author Response Figure 3 (For review only). Subcellular localization of OsSGS3a and OsSGS3b proteins.

Localization images of GFP-OsSGS3b or OsSGS3b-mCherry transiently expressed in the *ossgs3a-1* protoplast (upper) and GFP-OsSGS3a or OsSGS3a-mCherry transiently expressed

in the *oss3b* protoplast (lower). CFP served as a non-specific localized marker. Scale bars, 10 μ m.

Supplementary figure 8 needs statistical significance.

Re: Thanks for this kind suggestion. We analyzed the significant differences by One-way ANOVA with Tukey's HSD post hoc analysis ($P < 0.05$) in the new version of our manuscript. Although significant differences existed in the expression levels of *OsARF3a/3b/la* between NIP and *oss3b* mutants, or between NIP and *OsSGS3b-Flag* OE seedlings, it may be not associated with tasiR-ARFs. The results revealed that loss-of-function of *OsSGS3b* or overexpression of *OsSGS3a-Flag* or *OsSGS3b-Flag* may not consistently modulate the expression of *OsARF3a/b/la/lb* in leaves.

Supplementary figure 8 a-c shows variety in the different transgenic lines but #2 and #6 depict a big variation and the quantification numbers 0.75 and 1.05 don't seem to match with the variability. This could signify that the transgenic lines are not very uniform. In lines 226-229, the authors state there is no variability in the ARF overexpression, but there is variability in the expression of tasiR-ARFs.

Re: We thank the reviewer for these insightful comments. We re-performed the quantification of the intensity of blots and found that the abundance of tasiR-ARFs decreased in *OsSGS3b-Flag* OE#2 but remained unchanged in *OsSGS3b-Flag* OE#6 (Supplemental Fig. 8c). We repeated the RNA blot and qRT-PCR of tasiR-ARFs in the leaves of *OsSGS3b-Flag* OE seedlings. Both the results of RNA blot and qRT-PCR revealed that the expression levels of tasiR-ARFs were similar in different transgenic lines of *OsSGS3b-Flag* OE (Author Response Figure 4, for review only). We agree that the relative levels of tasiR-ARFs varied in different transgenic lines of *OsSGS3b-Flag* OE grown in the paddy field in independent experiments. This is likely induced by small fluctuations of environment and plant growth, and this probably reflects the true epigenetic nature of tasiR-ARFs production, other than stable control by genetic mechanisms. In the revised manuscript, we concluded our results as below: The loss-of-function of *OsSGS3b* or overexpression of *OsSGS3a/b-Flag* did not consistently

affect the abundance of tasiR-ARFs or the expression of *OsARF3a/b/la/lb* through the action of tasiR-ARFs in leaves (Supplemental Fig. 8a-f).

Author Response Figure 4 (For review only) The abundance of tasiR-ARFs in the leaves of *OsSGS3b-Flag* OE seedlings grown in the field.

a RNA blot of tasiR-ARFs. OsU6 was used as a loading control. The intensity of the blots was quantified.

b Levels of tasiR-ARFs as determined by qRT-PCR. OsU6 was used as an internal control. Significant differences were determined by One-way ANOVA with Tukey's HSD post hoc analysis ($P < 0.05$).

Figure 5: the labeling of positive and negative RNAi lines is confusing. What this means here. This figure needs a regular control like the WT to depict normal infection.

Re: Thank you for raising this point. The “positive” indicates the positive *OsSGS3* RNAi seedlings in which the expression levels of *OsSGS3a* and *OsSGS3b* are decreased when compared with the empty vector control *PTCK303* transgenic plants. The “negative” indicates the negative *OsSGS3* RNAi plants in which the expression levels

of *OsSGS3a* and *OsSGS3b* are similar as that of *PTCK303* transgenic plants. In the revised figure, we add the *PTCK303* empty vector transgenic plants as the control. In comparison with the *PTCK303* transgenic plants and negative *OsSGS3* RNAi plants, the positive *OsSGS3* RNAi plants displayed enhanced resistance to *M. oryzae* (Fig. 5a).

The supplementary figure 10 b infection profile is very variable. Were all spore concentrations equal throughout infections and was the same method for infection used? Some infections look like punch inoculations and others look like spray inoculation. The authors need to have consistent infections across all lines.

Re: Thanks for the comment. In the spraying inoculation assay, we used the same consistent spore concentration (1×10^5 spores/mL in sterile water containing 0.05% Tween-20) for all infections. The inoculation occasionally causes different lesion profiles with subtle different conditions or growth of seedlings. We re-performed the blast resistance assay of *oss3b* and *OsSGS3b-Flag* OE (Supplemental Fig. 10b), and *OsSGS3a-Flag* OE (Supplemental Fig. 10e) by spraying inoculation. Similar to the previous experiments, we did not detect consistently changed blast resistance in *oss3b*, *OsSGS3a-Flag* or *OsSGS3b-Flag* plants with both spray and punch inoculations (Supplemental Fig. 10b, e).

The authors state that reducing the expression of *OsSGS3* promotes more resistance to pathogens. However, figure 5 b shows less bacterial blight in the *OsSGS3* RNAi lines, but there is evident lesions present in the rice, consistent with fungal infection. What is this due to? Is this presenting bacterial blight infection or *M. oryzae* infection? This figure seems to show cross-contamination.

Re: Thank you for this insightful comment. Fig 5b showed typical bacterial leaf blight lesions but not blast lesions in the field-grown rice plants, which are completely different. Usually, field-grown rice plants will develop small brown spots that may be caused by naturally infecting *Bipolaris oryzae* in particularly with potassium deficiency. However, these brown spots are distinct from blast lesions which are spindle-shaped

with necrotic borders. We re-performed the bacteria blight resistance assay of *OsSGS3* RNAi plants and used the *PTCK303* transgenic plants as the control. In comparison with the *PTCK303* transgenic plants, the *OsSGS3* RNAi plants displayed enhanced resistance to *Xoo* (Fig. 5b).

This manuscript should provide a description of how bacterial and fungal infection looks like in rice. The infection phenotypes need to be established for readers to know how typical infections present in rice, especially when working with different phytopathogens.

Re: Thanks for this good advice. Blast lesions are spindle-shaped with necrotic borders. Upon *Xoo* infection, lesions usually develop as water-soaked to yellow-orange stripes on mechanically injured parts of leaves, and progress toward the leaf base. We have added the above description of symptoms of bacterial leaf blight and fungal blast in the “Pathogen inoculation and disease resistance assay” part in Methods.

Minor edits:

Figure legends should be consistent throughout all figures. Abbreviations should be included in the legend. For example, figure 1 should include what NT and HT mean.

Re: Thanks. We have carefully checked figure legends to make them consistent and we also explained all abbreviations used in the figures throughout the manuscript.

Figure 4: what is CK? Abbreviations should be included in the figure legend.

Re: CK in Fig. 4c and 4d indicates the corresponding seedlings grown at 28°C. We have changed it to 28°C.

Line 64: suppress to suppresses

Line 279: wild type misspelled

Re: Thanks for pointing these out. We have carefully checked the manuscript and corrected the typos and incorrect subject/verb associations.

REVIEWERS' COMMENTS

Reviewer #1 (Remarks to the Author):

Thanks to the authors for providing the additional experiments, especially immunoblot analysis of OsSGS3a in the leaves of wild-type inoculation with blast and bacteria treatment. The over-expression or suppression of OsSGS3 is able to show thermotolerance (in SGS3:FLAG OE) or disease resistance (in SGS3 RNAi) via the SGS3a-tasiRNA-ARF3 module. Therefore, the module may be useful for crop science in the future. However, in the wild-type plants, SGS3 protein levels are enriched after the treatment with Xoo (author response Figure1), and are decreased under the high-temperature treatments (Fig. 4e). When the authors discuss the wild-type role of the SGS3a-tasiRNA-ARF3 module with the SGS3 protein regulation under biotic and abiotic stress, it could be more supportive for our understanding. In the revised paper, the authors experimentally addressed all my questions.

Reviewer #2 (Remarks to the Author):

The authors have addressed all of my necessary revisions in this revised manuscript version. I have not further comments to be addressed.

Reviewer #3 (Remarks to the Author):

The authors have engaged with all of the issues I raised in my review of their original submission, and have answered them to my satisfaction in the revised manuscript. I am entirely satisfied with their responses and changes.

REVIEWERS' COMMENTS

Reviewer #1 (Remarks to the Author):

Thanks to the authors for providing the additional experiments, especially immunoblot analysis of OsSGS3a in the leaves of wild-type inoculation with blast and bacteria treatment. The over-expression or suppression of OsSGS3 is able to show thermotolerance (in SGS3:FLAG OE) or disease resistance (in SGS3 RNAi) via the SGS3a-tasiRNA-ARF3 module. Therefore, the module may be useful for crop science in the future. However, in the wild-type plants, SGS3 protein levels are enriched after the treatment with Xoo (author response Figure1), and are decreased under the high-temperature treatments (Fig. 4e). When the authors discuss the wild-type role of the SGS3a-tasiRNA-ARF3 module with the SGS3 protein regulation under biotic and abiotic stress, it could be more supportive for our understanding. In the revised paper, the authors experimentally addressed all my questions.

Re: We thank the reviewer for the very positive evaluation of our revised manuscript and the constructive comments on the manuscript. According to the reviewer's suggestion, we have added the following discussion about the role of the wild-type SGS3a-tasiRNA-ARF3 module with the SGS3 protein regulation under biotic and abiotic stress in L407: However, the abundance of OsSGS3a protein was slightly increased upon *M. oryzae* infection in wild-type plants, suggesting a differential modulation of OsSGS3 protein levels under biotic and heat stress. Importantly, we have showed that OsSGS3a and OsSGS3b enhanced rice thermotolerance while the suppression of OsSGS3a/b enhanced rice disease resistance. Therefore, the OsSGS3-tasiRNA-OsARF3 module is likely differentially regulated in face of abiotic and biotic stress.

Reviewer #2 (Remarks to the Author):

The authors have addressed all of my necessary revisions in this revised manuscript

version. I have not further comments to be addressed.

Re: We thank the reviewer very much.

Reviewer #3 (Remarks to the Author):

The authors have engaged with all of the issues I raised in my review of their original submission, and have answered them to my satisfaction in the revised manuscript. I am entirely satisfied with their responses and changes.

Re: We thank the reviewer for the appreciation of our revised manuscript.